

# In-situ measurements of trace gases, PM, and aerosol optical properties during the 2017 NW US wildfire smoke event

Vanessa Selimovic[1], Robert J. Yokelson[1], Gavin R. McMeeking[2], Sarah Coefield[3]

[1]Department of Chemistry, University of Montana, Missoula, 59812, USA
[2]Handix Scientific LLC, 5485 Conestoga Court, Suite 104B, Boulder, CO, 80301, USA
[3]Missoula City-County Health Department, Missoula, MT, 59801, USA

*Correspondence to:* R. J. Yokelson (bob.yokelson@umontana.edu)

**Abstract.** In mid-August through mid-September of 2017 a major wildfire smoke/haze episode strongly impacted most of the NW US and SW Canada. During this period our ground-based site in Missoula, MT experienced heavy smoke impacts for ~500 hours (up to 471 µg m$^{-3}$ hourly average $PM_{2.5}$). We measured wildfire trace gases, $PM_{2.5}$, and black carbon and sub-micron aerosol scattering and absorption at 870 and 401 nm. This may be the most extensive real-time data for these wildfire smoke properties to date. Our range of trace gas ratios for $\Delta NH_3/\Delta CO$ and $\Delta C_2H_4/\Delta CO$ confirmed that the smoke from mixed, multiple sources varied in age from ~2-3 hours to ~1-2 days. Our study-average $\Delta CH_4/\Delta CO$ ratio (0.166 ± 0.088) indicated a large contribution to the regional burden from inefficient "smoldering" combustion. Our $\Delta BC/\Delta CO$ ratio (0.0012 ± 0.0005) for our ground site was moderately lower than observed in aircraft studies (~0.0015) to date, also consistent with a relatively larger contribution from smoldering combustion. Our $\Delta BC/\Delta PM_{2.5}$ ratio (0.0095 ± 0.0003) was consistent with the overwhelmingly non-BC, mostly organic nature of the smoke observed in airborne studies of wildfire smoke to date. Smoldering combustion is usually associated with enhanced PM emissions, but our $\Delta PM_{2.5}/\Delta CO$ ratio (0.126 ± 0.002) was about half the $\Delta PM_{1.0}/\Delta CO$ measured in fresh wildfire smoke from aircraft (~0.266). Assuming $PM_{2.5}$ is dominated by $PM_1$, this suggests that aerosol evaporation, at least near the surface, can often reduce PM loading and its atmospheric/air-quality impacts on the time scale of several days. Much of the smoke was emitted late in the day suggesting that nighttime processing would be important in the early evolution of smoke. The diurnal trends show BrC, $PM_{2.5}$, and CO peaking in early morning and BC peaking in early evening. Over the course of one month, the average single scattering albedo for individual smoke peaks at 870 nm increased from ~0.9 to ~0.96. $B_{scat401}/B_{scat870}$ was used as a proxy for the size and "photochemical age" of the smoke particles with this interpretation being supported by the simultaneously-observed ratios of reactive trace gases to CO. The size/age proxy implied that the Ångström absorption exponent decreased significantly after about ten hours of daytime smoke aging, consistent with the only airborne measurement of the brown carbon (BrC) lifetime in an isolated plume. However, our results clearly show that non-BC absorption can be important in "typical" regional haze/moderately-aged plumes with BrC ostensibly accounting for about half the absorption at 401 nm on average for our entire data set.

## 1 Introduction

Biomass burning (BB) emissions are an important source of trace gases and particles that can influence local, regional, and global atmospheric chemistry, air quality, climate forcing, and human health (Crutzen and Andreae, 1990). BB is one of the largest sources of fine primary organic aerosol (OA), black carbon (BC), brown carbon (BrC) (Bond et al., 2004, 2013; Akagi et al., 2011), total greenhouse gases, and non-methane organic gases (NMOG) (Yokelson et al., 2008; 2009), which are precursors for the formation of ozone and OA. While the majority of BB occurs in the tropics, the small fraction of the global BB in the western US is responsible for a significant portion of US air quality impacts (Park et al., 2007; Liu et al., 2017) and contributes to increasing health concerns. Wildfire smoke has been shown to have adverse respiratory and cardiovascular health effects, is associated with mortality and morbidity, and exhibits lung toxicity and mutagenicity (Le et al., 2014; Liu et al., 2015; Reid et al.,



2016; Adetona et al., 2016; Kim et al., 2018). In some cases, long range transport of biomass burning emissions can cause air quality standards to be exceeded hundreds or thousands of kilometers downwind of the fire source (Jaffe et al., 2013; Wigder et al., 2013). In addition to health concerns, particulate matter from wildfires can reduce visibility, which can have impacts on safety and transportation (United States Environmental Protection Agency, 2016), and is a concern in protected visual

environments such as national parks and wilderness areas, much of which are in the western US, where a majority of wildfires occur. The Interagency Monitoring of Protected Visual Environments (IMPROVE) program initiated in 1985 implemented long term monitoring that establishes current visibility conditions and has helped to improve visibility in protected areas. However, record high temperatures, drought, and fire-control practices over the last century have culminated into a situation in which we can anticipate more frequent fires of a larger size and intensity in the Western US and Canada (Yue et al., 2015; Westerling et

al., 2006) that are expected to impact all aspects of air quality in the US—and have other impacts, including on visibility. In fact, over the last few decades, the annual number of wildfires in the US has not changed significantly, but the annual area burned has increased by a factor of about 3 (United States National Interagency Fire Center, 2017), and many of the highest burned-area years have coincided with many of the warmest years on record (United States Environmental Protection Agency, 2016). Despite these important issues, much of the emissions from BB remain either understudied or completely unstudied. To date, most of the

research on the emissions and evolution of smoke from US fires in the field has targeted prescribed fires (Burling et al., 2011; Akagi et al., 2013; Yokelson et al., 2013; May et al., 2014; Müller et al., 2016), and while there are studies that probe trace gas and optical property emissions of wildfire smoke sampled in the field (Liu et al. 2017; Lindaas et al., 2017; Landis et al., 2017; Collier et al., 2016; Eck et al., 2013; Sahu et al., 2012; Lack et al., 2012), much of the information is limited in temporal extent or incomplete chemically, and fails to assess important issues such as the aging and evolution of smoke over varying and

extended amounts of time, night time evolution and oxidation, or the contribution of constituents of increasingly recognized importance such as BrC (UV-absorbing OA), to name a few.

BrC emissions are typically mixed with co-emitted BC and non-absorbing OA, which can result in some measurement difficulties and uncertainty in isolating and evaluating the optical properties of BrC and its overall radiative impact (Wang et al., 2017). In lab-simulated wildfires, BrC was associated with smoldering combustion and accounted for up to 86% of absorption by

particles in the UV in the fresh smoke, which has several implications in atmospheric chemistry, including impacts on radiative forcing, UV-driven photochemical reactions producing ozone, and the lifetime of $NO_x$ and HONO (Selimovic et al., 2018). In addition, there are sources of BrC not directly emitted from BB, including the photo-oxidation of volatile organic compounds (VOCs) and aqueous-phase chemistry in cloud droplets. These processes produce BrC with optical properties and lifetimes that are not yet well-characterized (Graber and Rudich, 2006; Ervens et al., 2011; Wang et al., 2014; Laskin et al., 2015). In fact,

several factors such as chemical transformation, mixing state, combustion conditions, photochemical aging, etc., can all influence the absorption of BrC (Wang et al., 2017). Most modeling studies have found that despite the multiple variables contributing to the absorption of BrC, including BrC in climate models would mean the net radiative forcing of biomass burning would move in a more positive direction. (Feng et al., 2013; Jacobsen, 2014; Saleh et al., 2014; Forrister et al., 2015). Unfortunately, observational constraints on BrC are scarce making it difficult to assess and enhance models based on

observational evidence. Thus, more field measurements are required to get an accurate estimate of the impact of BrC both regionally and globally.

Most of the western US, including the Rocky Mountains, constitutes a large fire prone-region. Missoula, Montana is the largest city completely surrounded by the Rocky Mountains. Missoula is also located within a large region of the inland Pacific Northwest where wildfires have caused air quality trends to deviate from the pattern in the rest of the US (McClure and Jaffe,



2018). Missoula frequently experiences smoke impacts typical of much of the urban and rural west due to local and regional western fires. A few airborne studies have sampled western wildfires and are most sensitive to lofted emissions (Liu et al., 2017; Yates et al., 2016), but wildfires may produce some unlofted emissions, especially at night. Ground-based studies could probe these unlofted emissions, but have difficulty to representatively sample lofted emissions unless advection accompanies transport.

Despite these platform-based considerations, our laboratory on the eastern edge of Missoula is a relevant receptor for mixed-age (1-2 hours to 1-2 days) western wildfire smoke. In this study, we measured the wildfire smoke characteristics for 500 smoke-impacted hours during August-September of 2017, which constituted a prolonged period of record-breaking AQ impacts in Missoula. This very large sample of wildfire smoke helps address some of the afore-mentioned observational gaps in current wildfire field data. Specifically, two photoacoustic extinctiometers (PAXs), and a Fourier-transform-infrared spectrometer

(FTIR) characterized the smoke that entered the Missoula valley. A Montana Department of Environmental Quality (DEQ) $PM_{2.5}$ (particulate matter ≤2.5 micrometers in diameter) monitor provided additional insight and verified some impacts. The PAXs provided measurements of scattering and absorption at two wavelengths (nominal 405, actual 401 nm; 870 nm), BC, and derivations of single scattering albedo (SSA), and Angstrom absorption exponent (AAE) for $PM_{1.0}$. The FTIR measured the BB "tracer" carbon monoxide (CO) and a few other trace gases that help estimate "effective average smoke age", which can be

compared to changes in aerosol optical properties and inform model mechanisms. We present our results and compare them to those previously reported for wildfire field measurements and prescribed fire field measurements.

## 2 Experimental Details

### 2.1 Site Descriptions

Trace gases and particles were measured through co-located inlets at the University of Montana (UM), ~12.5 m above the ground

through the window of our laboratory on the fourth (top) floor of the Charles H. Clapp building (CHCB). The UM campus encompasses an area of ~0.89 $km^2$ and is located on the eastern edge of Missoula, with the CHCB located in the southeastern corner of campus. The CHCB is ~ 1.1 km from the nearest road that gets appreciable traffic during the summer, thus our measurements were not significantly influenced by automobile emissions (see Sect 3.1). $PM_{2.5}$ measurements were made by the Montana Department of Environmental Quality via a stationary $PM_{2.5}$ monitor located in Boyd Park, Missoula, which is ~3.2 km

southwest of our UM laboratory, with both sites being located in the Missoula valley proper.

### 2.2 Instrument Details

#### 2.2.1 Fourier transform infrared spectrometer

Trace gas measurements were made using an FTIR (Midac, Corp., Westfield, MA) with a Stirling cycle cooled mercury-cadmium-telluride (MCT) detector (Infrared Associates, Stuart, FL; Ricor USA Inc., Salem, NH) interfaced with a 17.22 m path

closed multipass White cell (Infrared Analysis, Inc., Anaheim, CA) that is coated with a halocarbon wax (1500 Grade, Halocarbon Products Corp., Norcross, GA) to minimize surface losses (Yokelson et al., 2003). Although the system was designed for source measurements, and is described elsewhere in more detail (Akagi et al., 2013; Stockwell et al., 2016a, Stockwell et al., 2016b), the FTIR is convenient for ambient monitoring because the Stirling cooled detector does not require refilling of liquid nitrogen and thus allows for mostly autonomous operation. Ambient air was drawn through the 2.47 liter White

cell at ~6 liters per minute via a downstream IDP-3 dry scroll vacuum pump (Agilent Technologies) using a 0.635 cm o.d. corrugated Teflon inlet that was positioned outside the window (~12.5 m above ground level). Cell temperature and pressure were also logged on the system computer (Minco TT176 TRD MKS Baratron 722A). Spectra were collected at a resolution of 0.50 $cm^{-1}$ covering a frequency range of 600-4200 $cm^{-1}$. A time resolution of approximately 5 minutes was more than adequate and sensitivity was increased by co-adding scans at this frequency. Gas phase species (with their respective detection limits in



parentheses), including carbon monoxide (CO, 20 ppb), methane ($CH_4$, 20 ppb), acetylene ($C_2H_2$, 2 ppb), ethylene ($C_2H_4$, 2 ppb), methanol ($CH_3OH$, 3 ppb), and ammonia ($NH_3$, 2 ppb) were quantified by fitting selected regions of the mid-IR transmission spectra with a synthetic calibration nonlinear least-squares method (Griffith, 1996; Yokelson et al., 2007). The uncertainties in the individual mixing ratios varied by spectrum and molecule and were influenced by uncertainty in the reference spectra (1-5%)

or the real time detection limit, whichever was larger. The procedure used to correct for gases outside of the spectrometer cell raised the uncertainty to ~20 ppb for background CO and $CH_4$, but did not affect the measured enhancements above background during smoke episodes. Calibrations with NIST-traceable standards indicate that peak CO values had an uncertainty of less than 5%. The FTIR system was designed for source sampling and the sensitivity was adequate to measure a significant amount of usable trace gas data, but not every species on every event. In addition, an FTIR system problem caused the trace gas data to

terminate about one day before the smoke cleared.

### 2.2.2 Photoacoustic extinctiometers (PAX) at 870 and 401 nm

Particle absorption and scattering coefficients ($B_{abs}$, $Mm^{-1}$, $B_{scat}$, $Mm^{-1}$) were measured directly at 1 s time resolution using two photoacoustic extinctiometers (PAX, Droplet Measurement Technologies, Inc., Longmont, CO; Lewis et al., 2008; Nakayama et al., 2015), and single scattering albedo (SSA) at 401 (nominally a 405 nm system) and 870 nm, and the Angstrom absorption

exponent (AAE) were derived using those measurements. Although the PAXs measured every second, data was averaged to 5 minutes, which was deemed adequate for the final analysis and matched the time resolution used by the FTIR for the same reason. A 1L $min^{-1}$ aerosol sample flow was drawn through each PAX using a downstream IDP-3 dry scroll vacuum pump (Agilent Technologies) and split internally between a nephelometer and photoacoustic resonator for simultaneous measurement of light scattering and absorption. Both PAX instruments contain an internal pump, however these internal pumps were bypassed

to improve measurement sensitivity, as the pumps can contribute an amount of acoustic noise that is noticeable in clean-air ambient measurements. Scattering of the PAX laser light was measured using the wide-angle (6º-174º) reciprocal nephelometer that responds to all particle types regardless of chemical makeup, mixing state, or morphology. For absorption measurements, the laser beam was directed through the aerosol stream and modulated at a resonant frequency of the acoustic chamber. Absorbing particles transferred heat to the surrounding air, inducing pressure waves that were detected via a sensitive microphone.

Advantages of the PAX include direct in-situ measurements, a fast response time, continuous autonomous operation, and eliminating the need for filter collection and the uncertainties that come with filter artifacts (Subramanian et al., 2007).

The PAX sample line was ~4.7 m of 0.483 cm o.d. conductive silicon tubing positioned outside the window ~12.5 m above ground level and co-located with the FTIR inlet. The tubing transferred outside air to a scrubber to remove light-absorbing gases (Purafil-SP Media, minimum removal efficiency 99.5%) and then a diffusion drier (Silica Gel 4-10 mesh) to remove water, with

post-drier relative humidity varying between 13 and 30%. The scrubber and drier were refreshed before any signs of deterioration were observed (e.g. color change) and the diffusion based designs should incur minimal particle losses, but losses were not explicitly measured. After the drier, a splitter connected to the two instruments. After the splitter, each sample line featured a 1.0 μm size-cutoff cyclone and two acoustic notch filters that reduced noise. Both PAX instruments were calibrated before, during, and after the experiment using the manufacturer-recommended scattering and absorption calibration procedures

utilizing ammonium sulfate particles and a propane torch to generate purely scattering and strongly absorbing aerosols, respectively. The 401 data was only used after August 27 because of frequent clogging of the $PM_{1.0}$ cyclone before that date. The estimated uncertainty in PAX absorption and scattering measurements has been estimated to be ~4-11% (Nakayama et al., 2015).

In the PAX, the incident laser light is absorbed in situ by light absorbing particles without filter or filter-loading effects that can be difficult to correct, particularly for samples with high organic aerosol loadings (Lack et al., 2008; Li et al., in prep). We





directly measure aerosol absorption ($B_{abs}$, Mm$^{-1}$) and used the literature- and manufacturer-recommended mass absorption coefficient (MAC) (4.74 ± 0.63 m$^2$ g$^{-1}$ at 870 nm) to calculate the BC concentration (µg m$^{-3}$) (Bond and Bergstrom, 2006), but the BC mass can be adjusted using different MAC values if supported by future work. Because the PAXs also measured light scattering, scattering and absorption values can be combined to directly calculate the single scattering albedo (SSA, the ratio of scattering to total extinction). SSA is a useful input for climate models, where an SSA closer to 1 indicates a more "cooling" highly-scattering aerosol:

$$SSA = \frac{Scattering\ (\lambda)}{Scattering\ (\lambda) + Absorption\ (\lambda)} \tag{1}$$

To a good approximation, sp$^2$-hybridized carbon (including BC) absorbs light proportional to frequency (Bond and Bergstrom, 2006). Thus, the $B_{abs}$ contribution from BC at 401 nm can be calculated from 2.17 times $B_{abs}$ at 870 nm (an absorption Angstrom exponent of one), where BrC absorption is expected to be negligible, and any additional $B_{abs}$ at 401 nm can be assigned to BrC ($B_{abs,\ BrC}$) subject to limitations due to "lensing" by coatings discussed elsewhere (Pokhrel et al., 2016; 2017; Lack and Langridge, 2013; Lack and Cappa, 2010). Coating effects are very difficult to isolate from BrC direct absorption effects and this adds some uncertainty to the BrC attribution (±25%), but not to the absorption measurements themselves. Additionally, the Ångström absorption exponent (AAE) (401/870) can be calculated from the 401 and 870 data, where the AAE of pure BC is usually close to one and larger values are indicative of smoke absorption more dominated by BrC emissions:

$$AAE = -\frac{\log\left(\dfrac{B_{abs,1}}{B_{abs,2}}\right)}{\log\left(\dfrac{\lambda_1}{\lambda_2}\right)} \tag{2}$$

The AAE is useful as an indicator of BrC/BC, but in addition, the full aerosol absorption spectrum is often approximated with a power law function (absorption = C × λ$^{-AAE}$) and thus the AAE determined with any wavelength pair can be used to approximately calculate the shape of absorption across the UV-VIS range (Reid et al., 2005b). An equation similar to equation 2 provides the Ångström scattering exponent (ASE), which can be used to calculate scattering at unmeasured wavelengths.

### 2.2.3 Montana Department of Environmental Quality PM$_{2.5}$

The Montana DEQ uses beta attenuation monitors (Met One Instruments, Model BAM-1020) in accordance with US EPA Federal Equivalent Methods (FEM) for continuous PM$_{2.5}$ monitoring. At the beginning of each sample hour, a constant $^{14}$C source emits beta rays though a spot of clean glass fiber filter tape. The beta rays are measured by a photomultiplier tube to determine a zero reading. The BAM-1020 then advances this spot of tape to the sample nozzle, where it filters a measured amount of outside air at 16.7 L/min. At the end of the sample hour, the attenuation of the beta ray signal by the filter spot is used to determine the mass (and concentration) of the particulate matter. Hourly detection limits for the BAM-1020 are <2.4 µg/m$^3$ (1σ). Current and archived air quality data for the state of Montana can be accessed using the following link: http://svc.mt.gov/deq/todaysair/. More information on the BAM-1020 can be found at http://metone.com/air-quality-particulate-measurement/regulatory/bam-1020/. Note PAX size cutoff was 1.0 micron and the PM size cutoff is 2.5 µm. The mass in the 1.0-2.5 µm range is thought to be a small part of the total mass (e.g. 10-20% in Fig. 2 in Reid et al., 2005a), but the size range





difference does affect data interpretation as detailed later. ($PM_{2.5}$ cyclones have now been obtained for the PAXs for ongoing studies.)

### 2.2.4 Emission ratios (ERs) and downwind enhancement ratios

We converted the time series of mixing ratios for each analyte measured into a form that is broadly useful to others for
implementation in local to global chemistry and climate models. To do this, we produce emission ratios (ERs) and enhancement ratios. The calculation of these two types of ratios is the same, but an emission ratio is only the appropriate term for a ratio measured directly at a source or further downwind for relatively inert species such as BC or CO. First, an excess mixing ratio (denoted by "$\Delta X$" for each species X) is calculated for all species measured by subtracting the relatively small background mixing ratio based on a sloping baseline from before to after a smoke impact. For example, the ratio for each species relative to
CO ($\Delta X/\Delta CO$) is the ratio between the sum of $\Delta X$ over the entire smoke impacted period relative to the sum of $\Delta CO$ over the entire smoke impacted period. Molar ratios to CO were calculated for BC, PM, and all the gases measured by the FTIR that exhibited enhancement above background levels for each smoke impacted period. Emission factors (EF), which can be derived by including the molar ER to $CO_2$ in the carbon mass balance method were not calculated (Selimovic et al. 2018). The diurnal variation for $CO_2$ is considerable, and the smoke was mainly aged (not reflecting initial emissions for most species) in Missoula.
The prolonged "small" $\Delta CO_2$ peaks that persist for times similar to the natural, substantial variation that $CO_2$ has have uncertain values. E.g., for $CO_2$, the wildfire smoke impacts in Missoula are largely diluted and protracted enough to not completely dominate background variability as is the case for the other gases and for source sampling (Stockwell et al., 2016a, Stockwell et al., 2016b, Akagi et al., 2011, Akagi et al., 2012). Since $\Delta CO_2$ are not as reflective of fire impacts, then by extension, the modified combustion efficiency (MCE) which is defined as $\Delta CO_2/(\Delta CO_2 + \Delta CO)$, is not as useful as an index of the combustion
flaming to smoldering ratio as in measurements closer to the source. Other approximate metrics of the relative amount of flaming to smoldering such as BC/CO or $CH_4$/CO can still be used.

### 2.3 Investigating smoke origin and back trajectory calculations

To investigate the sources contributing to smoke events we used a combination of back trajectory calculations, satellite imagery, and local meteorological data that provided insights into mixing and smoke origin. Back trajectories were calculated utilizing the
National Oceanic and Atmospheric Administration (NOAA) Air Resources Laboratory Hybrid Single Particle Lagrangian Integrated Trajectory (HYSPLIT; Stein et al., 2015; Draxler et al., 1999; Draxler et al., 1998; Draxler et al., 1997) initialized from UM (46.8601º N, 113.9852º W) at 500, 1200, and 3000 m above ground level during the hour at which enhancements for that particular smoke event were at a maximum. Back trajectories were run using the High Resolution Rapid Refresh (HRRR) operational model, which uses the uses the Weather Research and Forecasting (WRF) modeling system combined with
observational data assimilation and is run over the contiguous US at 3km × 3km resolution (Benjamin et al., 2016). For events that spanned multiple days, multiple back trajectories were initialized during the hour(s) at which enhancements for the sub-events were at a maximum. Because of the complex local topography and micrometeorology, the combination of back trajectories, satellite imagery (GOES "loops") and other evidence can only suggest a most likely smoke origin and cannot provide an exact smoke age. Our best guess at the smoke origin for each event is listed in Tab. S1.

### 2.4 Brief description of 2017 regional and selected local fires

Missoula experienced smoke impacts from local (western MT) and regional fires with regional fires including fires in California, Idaho, Oregon, Washington, and British Columbia. British Columbia experienced a record fire season, with over ~1.2 million ha





burned (BC Wildfire Service, 2017). More than 4 million ha burned in the US during the 2017 fire season, making it one of the largest to date. Idaho, Oregon, and Washington had burned areas over 263,000 ha, 283,000 ha, and 161,000 ha, respectively. California and Montana experienced their worst fire seasons to date, with both states experiencing close to 526,000 ha burned each (National Interagency Fire Center, 2017). Although the complicated meteorology and topography of the Missoula valley

makes attributing smoke sources somewhat difficult (as noted above), we can say with some degree of certainty that the majority of the fresh smoke impacting Missoula came from two local fires, the Lolo Peak fire and the Rice Ridge fire (Tab. S1). The Lolo Peak fire started at high elevation ~15 km SW of Missoula (46.674º N, 114.268º W) on 15 July 2017 and burned continuously (mostly at lower and lower elevations) until it eventually grew to over 20,000 ha. The fuel description as given by Inciweb (https://inciweb.nwcg.gov/incident/5375/) is summarized as containing generally sparse or patchy subalpine fir (*Abies*

*lasiocarpa*) with dead Whitebark pine (*Pinus albicaulis*) above ~2100 m. Below 2100 m, fuels were mainly typical of a variety of coniferous-dominated ecosystems with major tree species such as ponderosa pine (*Pinus ponderosa*), sub-alpine fir (*Abies lasiocarpa*), and lodgepole pine (*Pinus contorta*). Lower elevations near containment lines were dominated by ponderosa pine with grassy understory. The Rice Ridge fire started 24 July 2017 ~52 km NE of Missoula (47.268º N, 113.485º W). The fire eventually burned over 64,000 ha, with a notable run on 3 September 2017, where it doubled in size from ~20,000 ha to ~40,000

ha. Fuels involved were timber (litter and understory), and brush (https://inciweb.nwcg.gov/incident/5414/).

### 3. Results and discussion

### 3.1 Overview of 2017 fire season smoke impact in Missoula

Figure 1 shows the hourly average mixing ratios of CO, BC, and $PM_{2.5}$ observed from 11 August to 10 September 2017, which includes nearly all of the 2017 Missoula smoke impacts. There were more than 20 distinct periods of major smoke-impacts that

are readily identified by large simultaneous enhancements in CO, BC, and $PM_{2.5}$. The highest hourly values were observed on 4 September 2017, the morning after the Rice Ridge fire doubled in size ($PM_{2.5}$, 471 µg/m$^3$, CO 2.78 ppm, BC 3.62 µg/m$^3$). This event is discussed in more depth as a case study in a later section (3.5). Numerous other $PM_{2.5}$ peaks exceeded e.g. levels of 100 µg/m$^3$. "Cleaner" periods between smoke peaks became less extensive as the regional atmosphere became increasingly polluted until widespread clearing on 10 September 2017. Overall high correlation of CO and BC to $PM_{2.5}$ suggest that the smoke was

normally well mixed on the spatial scale that separated the $PM_{2.5}$ and UM monitors. Many of the longer smoke impacts that spanned several days were necessarily integrated as a single event for calculating ratios between species, but also probed as smaller "sub-events" to explore their source attribution, which could be mixed (Tab. S1).

### 3.1 Trace gas ratios

Table 1 reports study average ratios to CO for gases measured by the FTIR. These measurements are representative of

moderately aged regional wildfire smoke. We interpret our results by comparing them to emission ratios measured in the lab (Selimovic et al., 2018) and other field studies mostly in fresher smoke (Liu et al., 2017; Landis et al., 2017; Radke et al., 1991). CO is a major pollutant in the atmosphere with BB as a main source. In Missoula, especially in the summer, the CO background is not strongly influenced by non-fire sources. $CH_4$ on the other hand has more background variability, but at these smoke levels the ratio of $CH_4$ to CO, while variable, yields a study average (0.166 ± 0.088) that mostly reflects the real average $CH_4$/CO fire

emission ratio. Yates et al. (2016) reported a smoldering stage $CH_4$/CO ER of 0.095 (±0.023) for the Rim Fire, which is lower than our study average ER, but the ratio reported in Yates et al. (2016) comes from measurements closer to the source and from a single fire source. Our higher study average ER of $CH_4$ is indicative of smoldering, or specifically glowing combustion (Yokelson et al., 1997). Because the measurement was not in a direct downslope flow of smoke into Missoula, this ratio



suggests that smoldering emissions from regional fires can be and were frequently transported to the Missoula valley. This may be why our study average is higher than observed in airborne studies. In a consistent observation, we find that ERs for $CH_4/CO$ are lower when the BC/CO ERs are higher (Fig. 2), which is indicative of a flaming to smoldering ratio dependence. This is a useful result, because our two metrics for combustion characteristics at the fire source are consistent and it indicates that the

variability in ratios between species observed at Missoula was partly due to variable combustion types at the regional fire sources along with the expected effects of variable aging that are discussed next.

Next, we compare other measured trace gas ratios, including some more reactive VOC, to the limited amount of data available from previous airborne and lab studies. Liu et al. (2017) sampled smoke between 1-2 h old on average, and did not report an ER value for $NH_3$. However, Liu et al. (2017) reported an average wildfire MCE that Selimovic et al. (2018) used with

measurements of very fresh lab fire smoke to calculate an ER value for $NH_3/CO$ based on the average wildfire MCE reported in Liu et al. (2017). The predicted $NH_3$ value (0.0279) for wildfires based on an average wildfire MCE (0.91), is about twice our observed average $NH_3/CO$ (0.0133). Radke et al. (1991), measured an $NH_3/CO$ range from 0.037 for fresh smoke to 0.011 when including samples up to 48 h old. Our 2017 individual ratios span a range (Tab. S1). Near the high end we see $NH_3/CO$ of 0.0196 for relatively fresh smoke assigned to the nearby Lolo Peak Fire and 0.0216 for event "S" where the origin is unclear. Our lowest

ratios are about ¼ of our highest ratios (0.0044) (Tab. S1). Akagi et al. (2012) measured a mid-day $NH_3/CO$ half-life of ~5h, which suggests that our average sample age is roughly equivalent to ~5h of mid-day processing and our oldest samples are aged equivalent to about 10 hours of "mid-day processing" (Tab. S1). However, the "time since emission" is potentially longer since, according to the GOES satellite, a lot of smoke was produced in the evening and OH processing may not have started fully until the next day. In addition, we note that the true processing ages have potential to be even longer, since the true initial $NH_3/CO$

may have been higher than our highest observed ratios as we were not immediately adjacent to sources. This possibility is supported by the fact that $NH_3$ and $CH_4$ emissions have been shown to be linked (Yokelson et al., 1997), and our "high" $CH_4/CO$ value for event "S" (~0.14) could indicate that the real initial $NH_3/CO$ was higher than ~0.022. Finally, the $NH_3/CO$ ratio is also related to the size and age of particles as will be discussed in future sections (3.3).

$C_2H_4$ has been observed to decay in isolated plumes with a similar half-life to ammonia (Akagi et al., 2012; Hobbs et al., 2003),

and our study average $C_2H_4/CO$ ratio (0.011) is again about half that in the other wildfire studies reported in Tab. 1 (~0.02) or listed elsewhere (Akagi et al., 2011). Our lower $C_2H_4/CO$ ratios tended to occur when the $NH_3/CO$ ratio was also lower (Tab. S1), but unfortunately there are only two events with data for both gases and not enough measured values to warrant a detailed analysis. Methanol and acetylene react at least an order of magnitude more slowly with OH than $C_2H_4$. Our average methanol enhancement ratio (0.019) thus falls in the middle of the other wildfire values (0.0148 – 0.024) as might be expected when any

aging effects are smaller than the natural high variability in initial emissions (Akagi et al., 2011). In fact $CH_3OH/CO$ has been observed to increase or decrease slightly or stay the same (Akagi et al., 2012, Akagi et al., 2013, Müller et al., 2016). We have only a few data points for $C_2H_2/CO$, but their average is significantly lower than the other wildfire studies. Since $C_2H_2$ is associated with flaming combustion (Lobert et al., 1991; Yokelson et al., 2013) this could be due to the prevalence of smoldering that was also indicated by the high average $CH_4/CO$ ratios as noted above. Another point about our trace gas data is that our

mixing ratios for CO are valuable as an inert tracer for wildfire emissions for comparison to models and they can be useful for inferring the initial emissions of other gases when emission ratios to CO have been measured elsewhere (Selimovic et al., 2018; Koss et al., 2018; Liu et al., 2017). CO can also be used as a scaling/normalizing factor for particle emissions, which is discussed in the next section.

**3.2 $BC/PM_{2.5}$, BC/CO, $PM_{2.5}/CO$**





BC is estimated to be the second strongest global climate warming agent and BB is the main BC source. Accurate BC measurements are challenging and aerosol absorption remains poorly understood in atmospheric models (Bond et al., 2004; Bond et al., 2013). In contrast, CO is measured reliably at a network of surface sites and in aircraft campaigns, and can also be retrieved by satellite (MOPITT, IASI, AIRS, etc). As a result, CO emissions estimates are available for most sources, including

fires, and the estimates are in reasonable agreement for western wildfires (Liu et al., 2017). BC and BC/CO measurements by modern methods for wildfires are rare, thus, our BC, CO, and BC/CO measurements from a large sample of wildfire smoke can be used with CO emissions to update BC emissions estimates from wildfires (see below). BC is made only by flaming combustion at a fire source and despite the fact that its production rate can vary strongly with flame turbulence, the BC/CO ratio can serve as a rough indicator of the fire's flaming to smoldering ratio (Christian et al., 2003; Yokelson et al., 2009; Shaddix et

al., 1994) as exploited earlier in Fig. 2. Table 2 reports our study average ratios of BC/CO, BC/$PM_{2.5}$, and $PM_{2.5}$/CO and compares them to the limited measurements of wildfire smoke available in the lab (Selimovic et al., 2018) and in the field (Liu et al., 2017; Sahu et al., 2012; Hobbs et al., 1996). Our BC/CO ratio (0.0012) is a bit lower than the aircraft measured averages of Sahu et al. (2012) (0.0014), and Liu et al. (2017) (0.0016), and the Selimovic et al. (2018) estimate at the field average MCE for wildfires from Liu et al. 2017 (0.0018). The Hobbs et al. (1996) is notably higher than the other values and is actually an EC/CO

measurement that could be biased high. The Selimovic et al. lab average is also higher, but obtained at the higher lab-average MCE. The uncertainty in our value is likely asymmetric because coatings in aged PM could inflate absorption and our BC value by a small amount. Taken together, this suite of observations is consistent with our ground-based site being impacted by relatively more smoldering combustion compared to the other, mostly airborne, studies. Liu et al. (2017) calculated an annual CO production from western US wildfires of 5240 ± 2240 Gg, which they reported was in good agreement with an EPA estimate

from the 2011 National Emissions Inventory (4894). Ratioing to the Liu et al. estimate with the average field study BC/CO in Tab. 2 (0.0014 ± 0.0002) suggests that western US wildfires emit 7.3 ± 3.3 Gg of BC per year. This is significantly lower than a previous estimate, but the other estimate is not strictly comparable since it is based on EC measurements and for a different year (Mao et al., 2015).

Changes in the PM/CO ratio as a plume ages can be used as a metric for the net effect of secondary formation or evaporation of

organic and inorganic aerosol (Yokelson et al., 2009; Akagi et al., 2012; Jolleys et al., 2012; Vakkari et al., 2014). Table 2 indicates that our ground-based $PM_{2.5}$/CO (0.126 ± 0.002) is about half that obtained at aircraft altitudes in fresher wildfire smoke (0.266 ± 0.134) as reported by Liu et al. (2017) and ~4 times less than that reported for very fresh smoke by Hobbs et al., (1996) (0.492). Further our lower BC/CO ratio suggests enhanced smoldering, which should increase the PM/CO. Liu et al. (2017) and Forrister et al. (2015) measured smoke aging for the Rim Fire (a large California wildfire) as the plume aged and

found that the OA/CO ratio started high and then dropped to a value (0.125 ± 0.025) similar to our $PM_{2.5}$/CO. However, Collier et al. (2016) found no age dependence for OA/CO for plumes intercepted at Mount Bachelor or on the G-1 aircraft and obtained a value for OA/CO (0.25 ± 0.07) close to both the OA/CO and $PM_{1.0}$/CO of Liu et al. (2017). Taken together, these observations suggest that, on time scales up to ~1-2 days for the wildfire plumes studied to date, aging and/or higher average temperatures at lower elevations may encourage some OA evaporation and reduce downwind PM impacts, Some studies in other fire types have

found secondary formation to dominate at low elevation (Yokelson et al., 2009; Vakkari et al., 2014) so it is premature to generalize this observation to all BB and more study is needed. However, both of the latter studies measured smoke within a few hours of the source, and our lower $PM_{2.5}$/CO indicates that evaporation of PM dominated over formation of PM as smoke was transported to the Missoula valley in smoke that was between several hours and several days old.



The climate impacts of smoke are strongly related to the BC/PM ratio and also the SSA and BrC, which are described in more detail in other sections. The BC/PM ratio also allows for an estimate of ambient BC from ambient PM data when BC isn't measured, but caution is needed since PM may not be conserved as long as BC. Our average $BC/PM_{2.5}$ ratio (0.0095, Fig. 3) is higher than the average $BC/PM_{1.0}$ in Liu et al. 2017 (0.006) but falls within the range observed for two wildfires measured in Liu

et al. (2017), despite the differences in measurement techniques ($PM_{2.5}$ vs. $PM_{1.0}$, etc). It's possible that the BC/PM ratio reported in this study is up to ~30% too high if we consider the effects of coating on BC and lensing as a positive error (Pokhrel et al., 2017). A previous study found that smoldering combustion emits anywhere between 4-49 times more PM than flaming combustion (Kim et al., 2018), so the combination of our BC/CO ratio that is indicative of more smoldering combustion and a BC/PM ratio that is similar to or slightly above measurements closer to fire sources (Liu et al., 2017) again suggests that some

net evaporation of PM is occurring between the wildfire sources and our surface site. Again, this is worth more study since this could modify air quality and health effects.

OA is the main component of PM and the BC/PM ratio is likely similar to the BC/OA ratio. Our BC/PM ratio (~1%) then suggests that the aerosol measured was overwhelmingly organic, and thus strongly cooling, especially if the impact of BrC or lensing was small. Further, the mass-absorption coefficient (MAC) for OA scales with the BC/OA ratio (Saleh et al., 2014) so we

anticipate a low MAC, which is explored more next.

### 3.3 UV-absorption by brown carbon

While the attribution of BrC is not exact and varies across studies (Pokhrel et al., 2017), BrC absorption will offset the climate cooling calculated for purely-scattering OA depending on the amount emitted, its MAC, and its lifetime (Feng et al., 2013). One field study of BrC lifetime suggests a significant decrease of BrC over the course of a day, but a prolonged persistence of BrC

nonetheless (~6% above background even after 50h following emission) (Forrister et al., 2015), and studies of relevant chemical mechanisms involving BrC have shown both increases and decreases (Lin et al., 2015; Liu et al., 2016; Xu et al., 2018). Satellite retrievals employing reasonable a-priori aerosol layer heights indicate that BrC can have a strong impact in fresh BB plumes and a persistent significant impact in downwind regional haze (Jethva et al., 2011; Hammer et al., 2016). Here we present in-situ data showing persistent widespread regional impacts of BrC. Table 3 lists the study-average AAE and percent contribution to

absorption at 401 nm by BrC. We interpret our results by comparing them to the limited measurements of wildfire smoke in the lab and field and measurements for "flaming dominated" savanna fires (Selimovic et al., 2018; Forrister et al., 2015; Eck et al., 2013). Theoretically, aerosol absorption that is dominated by black carbon would have an AAE close to 1.0 (Bergstrom et al., 2002; Bond and Bergstrom, 2006; Bergstrom et al., 2007), which is the case in Eck et al., 2013 where they report an average AAE of 1.20 for measurements of savannah fires in southern Africa. On the other hand, Selimovic et al. (2018) and Forrister et

al. (2015) calculated AAEs for fresh smoke of 3.31 and 3.75, respectively, for various mixed coniferous fuels burned in a laboratory and in the field. Our study average AAE (1.96 ± 0.38) is almost 2 times lower than the average value recommended for fresh wildfire smoke (~3.5) in Selimovic et al. (2018), but higher than that reported in Eck et al. (2013). This is also the case for the percent contribution to absorption at 401 nm by BrC, where a lower AAE corresponds to lower BrC absorption. The AAE recommended for fresh wildfire smoke implies the %-absorption by BrC at 401 nm is close to 86%, but we still see significant

(~50%) absorption by BrC at 401, on average, in our moderately aged smoke.

Although we cannot determine precise smoke ages in this study, we can construct an analysis of our data that probes the trend in AAE and % absorption by BrC with aging. We start by noting that Mie scattering calculations (J. Walker, personal communication, 2017) imply that the ratio of $B_{scat}401/B_{scat}870$ should decrease as average particle size increases (e.g. Schuster et



al., 2006; Eck et al., 1999; Kaufman et al., 1994) and average particle size is well-known to increase with particle age (Akagi et al., 2012; Eck et al., 2013; Carrico et al., 2016). We also show in Fig. 4a that the $NH_3/CO$ ratio decreases with $B_{scat}401/B_{scat}870$ and we know $NH_3/CO$ decreased with aging with a ~5 hour half-life in the fall and under slower photochemical conditions in Tab. 2 in Akagi et al. (2012). Thus, the range in $B_{scat}401/B_{scat}870$ shown in Fig. 4a represents about 10 hours of day-time aging.

We also see a weak trend, but significant decrease in AAE over a similar range of our size/age parameter in Fig. 4b. Our data for AAE versus a proxy for aging time for multiple plumes is more variable than the AAE versus known transport time for a single plume in Forrister et al. (2015), but still supports a similar conclusion: that the net effect of BrC aging is a substantial decrease in AAE over the course of ~10 hours of aging.

       We also speculate that, in addition to aging, the time of day that smoke is formed may impact BrC and AAE. We motivate that

hypothesis next and then explore the issue in subsequent sections. Selimovic et al. (2018) showed that BrC accounted for most of the absorption at 401 nm when MCEs were in a low range associated with dominant smoldering combustion. Benedict et al. (2017) further observed that smoke impacts from a nearby wildfire had a much higher smoldering/flaming ratio at night than during the day, which then suggests the potential for increased BrC formation at night. It is also known that smoldering combustion of biomass emits many precursors, including monoterpenes, furans, cresol, etc. (Stockwell et al., 2015); that can

react quickly with the major night time oxidant, $NO_3$, and ostensibly form UV-absorbing organic nitrates that could augment BrC. In fact, estimates using current data strongly suggest that a substantial nighttime secondary BrC source could exist. The EF for primary organic aerosol (POA) produced by BB typically ranges from 3 to 30 g/kg (May et al., 2014; Liu et al., 2016, 2017). The EF for known plus unidentified non-methane organic gases (NMOGs) with intermediate to low volatility ranges from 3 to 100 g/kg. Converting even a small percentage of the co-emitted NMOGs that are known to react quickly with $NO_3$ could yield

substantial amounts of BrC and build up a reservoir of BrC during dark hours. Once daytime commences, other studies show that some types of BrC, depending on the precursor, can experience rapid photochemical degradation or formation via both direct photolysis and oxidation (Zhao et al., 2015; Lee et al., 2014, Zhong and Jang et al., 2014; Sareen et al., 2013). In summary, our extensive in-situ measurements show that even after 1-2 days of aging, BrC remains a significant component of ambient smoke, and that the climate properties of the regional haze have a non-BC absorption contribution. However, the details of the formation

and lifetime of BrC are complicated and probably vary diurnally.

### 3.4 Single Scattering Albedo, Mass Absorption Coefficient, Mass Scattering Coefficient

       This section starts with an important reminder/caveat. Our scattering and absorption data is measured for particles up to 1.0 μm, but the PM mass reported by the Missoula DEQ site includes particles up to 2.5 μm. Thus, using our data to calculate mass absorption coefficients (MAC) and mass scattering coefficients (MSC) will produce lower limit values that are not directly

comparable to those obtained when the range for both optical and mass measurements goes up to 2.5 μm. Nevertheless potentially useful to link $PM_{1.0}$ and $PM_{2.5}$ measurements since measurements at 1 μm cutoffs are common in field campaigns, but $PM_{2.5}$ still remains the common measurement in regional networks.

       Our MAC and MSC values were calculated by plotting 1-hr averages of $B_{scat}401$, $B_{abs}401$, and $B_{scat}870$, $B_{abs}870$ versus the 1-hr $PM_{2.5}$ values to calculate an MSC(401), MAC(401), MSC(870), MAC(870), respectively (Fig. S1). Values at other wavelengths

were calculated using a linear regression using the calculated averages. Our $(PM_{1.0}/PM_{2.5})$ MSC values are lower than those reported for $PM_{2.5}/PM_{2.5}$, but still potentially useful. For instance, the $PM_{1.0}/PM_{2.5}$ MSC at 870 nm is unity to a good approximation, which suggests a convenient way to estimate $PM_{2.5}$ directly from PAX-870 scattering data. Using a 1-micron cut-off probably isolated the combustion-generated OA and BC pretty well, but dust, ash and biological particles can be physically




entrained in wildfire plumes (Formenti et al., 2003; Gaudichet et al., 1995; Hungershoefer et al., 2008). The particles in the 1.0-2.5 micron range are a small part of the total mass in smoke emissions (Reid et al., 2005a) but they contribute disproportionately to the scattering. The additional absorption that we might have measured with a 2.5 micron cutoff may be less significant. Our study average MAC at 401 nm is only $0.19 \pm 0.08$ m$^2$ g$^{-1}$, consistent with a low BC/OA ratio (Saleh et al., 2014).

SSA, AAE, and ASE are commonly used to calculate aerosol absorption and scattering in models and satellite retrievals. (Ramanathan et al., 2001; McComiskey et al., 2008). Uncertainty in the SSA is one of the largest sources of uncertainty in estimating the radiative effect of aerosols (Jiang and Feingold, 2006; McComiskey et al., 2008). Some models and satellite (e.g. MODIS) retrievals assume a constant value of SSA for fire aerosol throughout the biomass burning season and the entire year, which may be an inaccurate approach. Eck et al. (2013) found an increase in SSA at 550 nm from 0.81 in July to 0.88 in October

in southern Africa. In Fig. 5 we present evidence for an increase in the SSA for moderately aged wildfire plumes over a prolonged period of biomass burning. While we did not directly measure SSA at 550 nm, we did measure SSA at 870 nm for the duration of the sampling period and SSA at 401 nm for the duration that the PAX 401 was operational. Figure 5 shows a moderate increasing trend in the SSA at 870 nm, but no significant trend in the SSA at 401 nm. It could be that because the sampling period of the PAX 401 nm only covers ~2 weeks, any trend that may be present is not apparent within this time frame.

Table 2 in the supplement shows our study average SSA at 870 nm and 401 nm, both of which are ~0.93, which is similar to the SSA reported at 550 nm in McMeeking et al. (2005b) of 0.92. Our SSA and the SSA reported in McMeeking et al. (2005b) are higher than the sometimes quoted typical surface SSA of the earth (~0.9, Praveen et al., 2012) which suggests that the wildfire PM$_{1.0}$ in regional haze would contribute to regional cooling (Thornhill et al., 2018; Kolusu et al., 2015). Conversely, an SSA range like that reported in Eck et al. (2013) could contribute to warming, which could potentially contribute to a positive-

feedback cycle associated with biomass burning (Jacobsen, 2014).

### 3.5 Case Study: Labor Day Weekend

Figure 6 highlights our data for Labor Day weekend (LDW), spanning ~50 hours from 4 September 2017 to 5 September 2017. We focus on this time period because it includes the largest impacts in Missoula, a regional smoke-production episode detected as far downwind as Europe (Ansmann et al., 2018), and an opportunity to compare what is likely smoke from one fire, subjected

to different processing scenarios. Peak "V" is smoke that was likely primarily produced at night and transported to Missoula at night before subsequent photochemistry and dilution in the Missoula Valley. In contrast, peak "W" is smoke that was likely produced and transported during the day before aging in Missoula. Surface winds observed coming from the east, our back trajectory calculations, and satellite observations along with the high concentration values of peak V all imply that the smoke was mostly sourced from a local fire (Rice Ridge). Our peak-integrated proxy for particle size (4.02, smaller particle size) and

the peak-integrated NH$_3$/CO ratio ($9.66 \times 10^{-3}$) for peak "V" suggest that the smoke retained fairly fresh characteristics even factoring in the daytime tail on the peak (Tab. S2). The peak integrated AAE (2.88) is the highest observed value for AAE from this study for any peak where an AAE could be derived. The same is true for the %401-absorption by BrC (~77%). The UV absorption results are within the range observed for fresh smoke reported in Selimovic et al., 2018 and reiterated again earlier in Tab. 3, which lists average AAE values for fresh smoke between 2.80 and 3.75 (Forrister et al., 2015). Average values for %401-

absorption by BrC in fresh smoke ranged between 64 and 86% (Selimovic et al., 2018), and again our integrated result for peak V falls in this range. In summary, the moderately-aged, strongly night-influenced peak has properties not inconsistent with significant amounts of BrC due to smoldering combustion or substantial nighttime BrC formation via reactions with NO$_3$ or O$_3$.



While not readily apparent via satellite observations due to stacked smoke layers, our back trajectory calculations, a similar peak shape on an upwind monitor, visual observations of a wall of smoke arriving from the northeast, and high concentrations of PM at the Missoula measuring site strongly suggest that peak "W", with an onset in the early evening, also mostly came from the Rice Ridge Fire as daytime produced/processed smoke. Peak "W" has a 401/870 scattering ratio (2.65) that implies larger

particle sizes and an $NH_3/CO$ ratio (0.0044) that is ~50% that of Peak "V". The ratio of $C_2H_4/CO$ decreases by ~30% from peak V to peak W. The AAE for peak "W" is 2.00, which is ~30% less than the AAE for Peak "V", and corresponds to a lower %401-absorption by BrC for the evening-onset peak (~54%). Taken together, these values imply larger particles and more photochemically aged smoke. Interestingly, the ratio of $CH_4/CO$ and $BC/CO$ are essentially similar for peaks V and W. This implies the flaming/smoldering ratio at the source for these events was similar ($NO_3$ chemistry could still have been more

important for peak V). While nighttime wildland fire combustion may be normally more smoldering dominated, LDW was marked by an unusual lack of nighttime RH recovery and an aggressive doubling of the fire size. Thus data from a different, more typical period is likely needed to probe diurnal differences in fresh smoke, which we examine next.

### 3.6 Diurnal Cycles

Diurnal cycles of smoke measured in Missoula provide some insight into regional meteorological effects and have some

potential to further probe the day versus night flaming/smoldering issues raised in the previous section (3.5). There is, however, a variable delay from production to receptor. Figure 7 shows the diurnal cycle of CO and the average hourly $PM_{2.5}$ measured across the entirety of the smoke sampling period. Levels of CO and $PM_{2.5}$ peak together from about 5 to 11AM, which is consistent with increased smoldering at night, but would also reflect the mixed layer height. Figure 8 shows the diurnal cycles of $PM_{2.5,}$ hourly average BC, and hourly average %401-absorption by BrC. In this case we see that "potential" BrC absorption

peaks in the early AM while BC peaks in the evening. One possible explanation for this is that despite variation in mixed layer height there is "typically" an increase in the flaming to smoldering ratio that produces more black carbon during the day. However, we can't rule out that an increase in photo-bleaching throughout the middle of the day impacts the peak position for absorption by BrC, but even then, the absorption by BrC remains about half of the absorption at 401 nm.

### 3.7 Brief comparison to prescribed fire data

Of the 718 hours we sampled during August and September 2017, 500.5 hours were part of a smoke event, which is close to three quarters (~70%) of the total monitoring time period. Of the total 718 hours of monitoring, over half (56%) violated the National Ambient Air Quality Standards (NAAQS) for allowable $PM_{2.5}$ averaged over 24 hours (35 µg/m$^3$). The hourly average for the entire sampling period of ~54 µg/m$^3$ of $PM_{2.5,}$ is an average exceedance of the 24-hour NAAQS standard by 42%. One possible approach to minimizing wildfire AQ impacts is pre-emptive prescribed burning. Prescribed fires reduce hazardous fuels,

burn less fuel per unit area, make less smoke per unit fuel consumption, and allow controlled dispersion conditions (Liu et al., 2017).

It is of interest to compare our large sample of ambient wildfire data to the comparatively rare data from airborne wildfire studies and prescribed fire data to see if our large sample size supports the earlier (Liu et al., 2017) conclusions regarding the nature of the smoke and emissions. More strongly supported conclusions can reinforce the land management implications. Table 5 lists

the BC/CO, BC/PM, and PM/CO ratios for our ambient wildfire study, the airborne wildfire study from Liu et al., 2017, and prescribed fire values reported in May et al., 2014. The available PM/CO data for wildfires is consistently higher than for prescribed fires, which has air quality and land management implications.



The available PM/BC ratios are consistently ~20 times higher for wildfires, than prescribed burns, confirming that wildfire smoke is overwhelmingly more organic, which is important partly because many optical properties scale with the BC/OA ratio. In general, our ground-based wildfire study confirms the earlier airborne indications that prescribed fires are less smoky but also less cooling than wildfires.

**4 Conclusions**

A major, prolonged wildfire smoke/haze episode impacted the NW U.S. and SW Canada during August through September of 2017. During this episode, we collected over 500 hours of data characterizing smoke/haze properties with FTIR and PAXs at 870 and 401 nm at a ground-based site in Missoula, MT. This is probably the most extensive real-time data on wildfire smoke properties to date. Our low BC/PM ($0.0095 \pm 0.0005$) ratio confirmed the overwhelmingly organic nature of the smoke observed

in the airborne studies of wildfire smoke to date. Our BC/CO ratio ($0.0012 \pm 0.0005$) for our ground site was moderately lower than observed in aircraft studies suggesting a relatively larger contribution from smoldering combustion. Despite our lower BC/CO ratio our PM/CO ratio was about half that measured in fresh smoke from aircraft. This suggests that OA evaporation, at least near the surface, may typically reduce PM air quality impacts on the time scale of several days. $B_{scat}401/B_{scat}870$ was used as a proxy for size and age of the smoke particles with this interpretation being supported by the trace gas data. The size/age

proxy implied that AAE decreased significantly after about ten hours of smoke aging, consistent with the single BrC lifetime measurement in an isolated plume. The results clearly show that non-BC absorption can be important in "typical" regional haze/moderately-aged plumes with BrC accounting for about half the absorption at 401 nm on average for the entire data set. The diurnal trends show BrC, PM, and CO peaking in early morning and BC peaking in early evening. Over the course of one month, the SSA at 870 nm increased from ~0.9 to ~0.96.

**Data Availability**

Raw data used to derive ERs and other quantities reported that are not included in the supplemental information can be obtained by contacting the corresponding author.

**Author Contributions**

VS and RY conducted the research and/or contributed to the data analysis. All authors contributed to the discussion and

interpretation of the results and writing the manuscript.

**Competing Interests**

The authors declare that they have no competing interests.

*Acknowledgements*

Vanessa Selimovic and Robert Yokelson were supported by the NSF grants AGS-1748266 and AGS-1349976, NOAA-CPO

grant NA16OAR4310100, and NASA grant NNX13AP46G to UM. Gavin McMeeking was supported by the NOAA-CPO grant NA16OAR4310109. Purchase and preparation of the PAXs was supported by NSF grant AGS-1349976 to R. Y. We thank John Walker for providing us with Mie scattering calculations.



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



Table 1. Study average enhancement ratios (ratioed to CO) compared to emission ratios reported in other studies.

| Compounds | This Work | Selimovic et al., 2018[a] | Selimovic et al., 2018[b] | Liu et al., 2017 | Landis et al., 2017 | Radke et al., 1991[c] |
|---|---|---|---|---|---|---|
| Methane (CH$_4$) | 0.1661 (0.0884) | 0.0741 (0.0698) | 0.0870 | 0.0960 (0.0425) | 0.104 (0.001) | 0.0503 (0.0420) |
| Acetylene (C$_2$H$_2$) | 0.0014 (0.0004) | 0.0062 (0.0607) | 0.0056 | 0.0028 (0.0022) | -- | 0.0023 (0.0018) |
| Ethylene (C$_2$H$_4$) | 0.0114 (0.0022) | 0.0209 (0.0193) | 0.0199 | 0.0102 (0.0033) | -- | -- |
| Methanol (CH$_3$OH) | 0.0199 (0.0013) | 0.0148 (0.0152) | 0.0176 | 0.0240 (0.0160) | -- | -- |
| Ammonia (NH$_3$) | 0.0133 (0.0064) | 0.0232 (0.0350) | 0.0279 | -- | -- | 0.0219 (0.0099) |

[a]Measured lab values at lab fire MCE

[b]Calculated from EF vs MCE fit based on average wildfire MCE reported in Liu et al.

[c]Averages of Myrtle Fall Creek and Silver Fire



Table 2. Study average enhancement ratios (ratioed to CO) compared to emission ratios reported in other studies.

| Ratios | This Work | Selimovic et al., 2018[a] | Selimovic et al., 2018[b] | Liu et al., 2017[c, d] | Sahu et al., 2012 | Hobbs et al., 1996[e] |
|---|---|---|---|---|---|---|
| BC/CO | 0.0012 (0.0005) | 0.0087 | 0.0018 | 0.0016 (0.0018) | 0.0014 | 0.0103 |
| BC/PM$_{2.5}$ | 0.0095 (0.0003) | -- | -- | 0.0060 (0.0054) | -- | -- |
| PM$_{2.5}$/CO | 0.1263 (0.0015) | -- | -- | 0.2661 (0.1342) | -- | 0.4923 |

[a] Measured lab values at lab fire MCE

[b] Calculated from EF vs MCE fit based on average wildfire MCE reported in Liu et al.

[c] Average of Rim Fire and Big Windy Complex. BC data was analyzed for Liu et al. (2017) study, but not reported.

[d] PM values reported are PM$_{1.0}$

[e] PM values reported are PM$_{3.5}$




Table 3. Study average AAE & %BrC contribution compared to other studies.

|  | This Work | Selimovic et al., 2018a | Selimovic et al., 2018b | Forrister et al., 2015 | Eck et al., 2013 |
|---|---|---|---|---|---|
| AAE | 1.96 (0.38) | 2.80 (1.57) | 3.31 | 3.75 | 1.20 |
| %BrC | 50.72 (12.78) | 64.19 (17.20) | 78.00 | -- | -- |

[a] Measured lab values at lab fire MCE

[b] Calculated from average wildfire MCE reported in Forrister et al., 2015.



Table 4. Study average SSA, MAC, and MSC compared to other work.

| Parameter | λ (nm) | This Work | Selimovic et al., 2018[b] | Selimovic et al., 2018[c] | Eck et al., 2013 | McMeeking et al., 2005 | Reid et al., 2005b |
|---|---|---|---|---|---|---|---|
| SSA | 401 | 0.93 (0.01) | 0.79 (0.13) | 0.9 | -- | -- | -- |
| | 540 | 0.933[a] | -- | -- | -- | -- | 0.85 (0.03) |
| | 550 | 0.933[a] | -- | -- | 0.81-0.88 | 0.92 (0.02)[d] | 0.86-0.90 |
| | 870 | 0.94 (0.02) | 0.64 (0.26) | 0.92 | -- | -- | -- |
| MAC | 401 | 0.23 (0.01) | -- | -- | -- | -- | -- |
| | 530 | 0.178 | -- | -- | -- | 0.37 (0.05)[e] | -- |
| | 540 | 0.174 | | | | | 0.7 (0.4) |
| | 550 | 0.170 | -- | -- | -- | -- | 0.7-0.8 |
| | 870 | 0.04 (<0.01) | -- | -- | -- | -- | -- |
| MSC | 401 | 3.23 (0.06) | -- | -- | -- | -- | -- |
| | 530 | 2.62 | -- | -- | -- | 5.5 (0.5)[e] | -- |
| | 540 | 2.57 | -- | -- | -- | -- | 3.2-4.2 |
| | 550 | 2.52 | -- | -- | -- | -- | 3.6-3.8 |
| | 870 | 1.01 (0.02) | -- | -- | -- | -- | -- |

[a]Calculated values using fit based on 401 and 870 nm values.
[b]Measured values at lab fire MCE.
[c]Calculated from EF vs MCE fit based on averaged wildfire MCE reported in Liu et al., 2017.
[d]McMeeking et al., 2005b
[e]McMeeking et al., 2005a



Table 5. Comparison of wildfire enhancement ratios to prescribed fire emission ratios

| Ratios | This Work | Liu et al., 2017[a, b] | May et al., 2014[b] |
|---|---|---|---|
| BC/CO | 0.0012 (0.0005) | 0.0016 (0.0018) | 0.013 (0.007) |
| BC/PM2.5 | 0.0095 (0.0003) | 0.0060 (0.0054) | 0.163 (0.019) |
| PM2.5/CO | 0.1263 (0.0015) | 0.2661 (0.1342) | 0.080 (0.030) |

[a] Average of Rim Fire and Big Windy Complex. BC data was analyzed for
Liu et al. (2017) study, but not reported.
[b] PM values reported are PM1.0



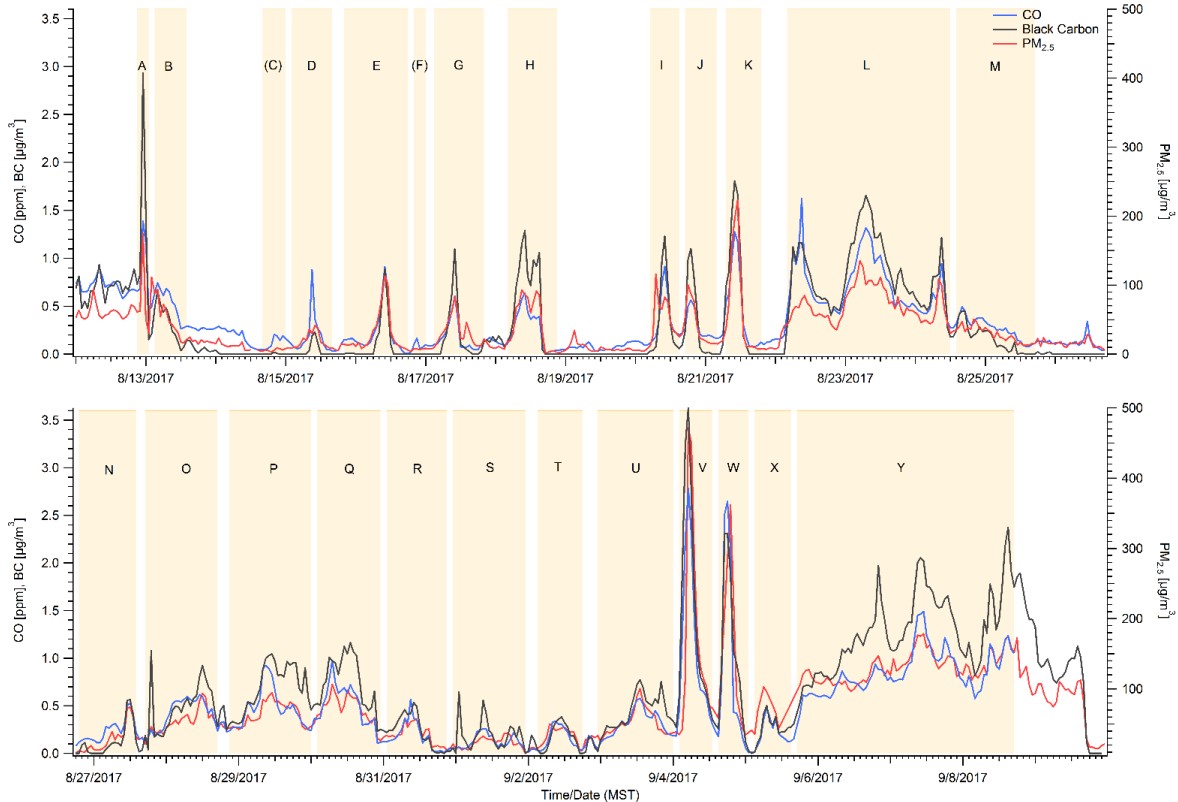

Figure 1. Time series of hourly CO, BC, and PM$_{2.5}$ measurements from Missoula. Sections highlighted in yellow indicated smoke-impacted periods. Peaks labeled with a parentheses indicated events that could not be attributed to biomass burning sources, and were excluded from analysis.





Figure 2. Methane emission ratio versus black carbon emission ratio. Point shown are for events that have both a CH$_4$/CO ratio and a BC/CO ratio.

Figure 3. BC/PM ratio.





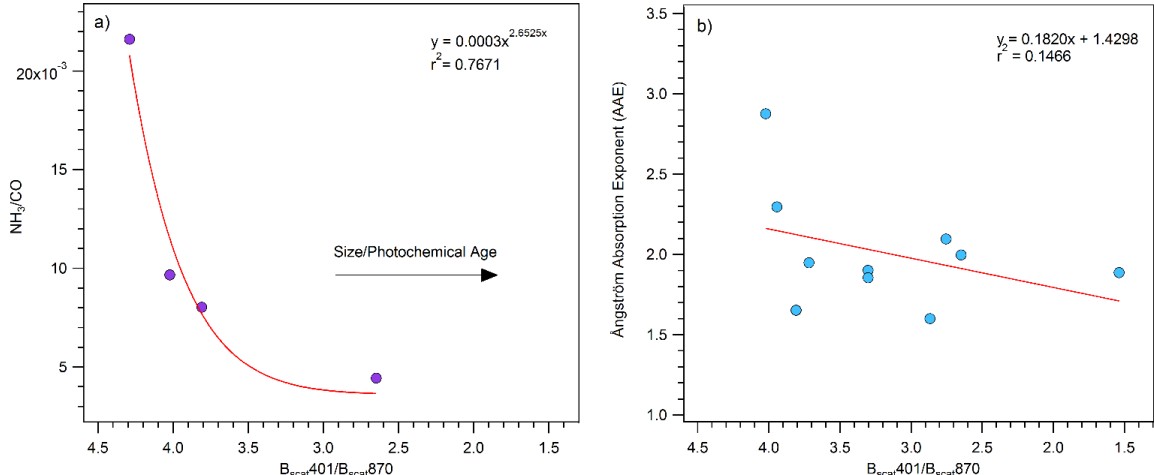

Figure 4: a. Plot of our size proxy (401 Scattering/870 Scattering) versus the Angstrom absorption exponent. b. Plot of our size proxy (401 Scattering/870 Scattering), versus the NH₃/CO ratio. Points shown in both graphs are at 1-hr time resolution for smoke impacts that have an NH₃/CO ratio and size proxy (when both PAXs were operational).







Figure 5. Plot of single scattering albedo over the course of the ambient smoke-monitoring period. Points represent SSA from smoke-impacted events.



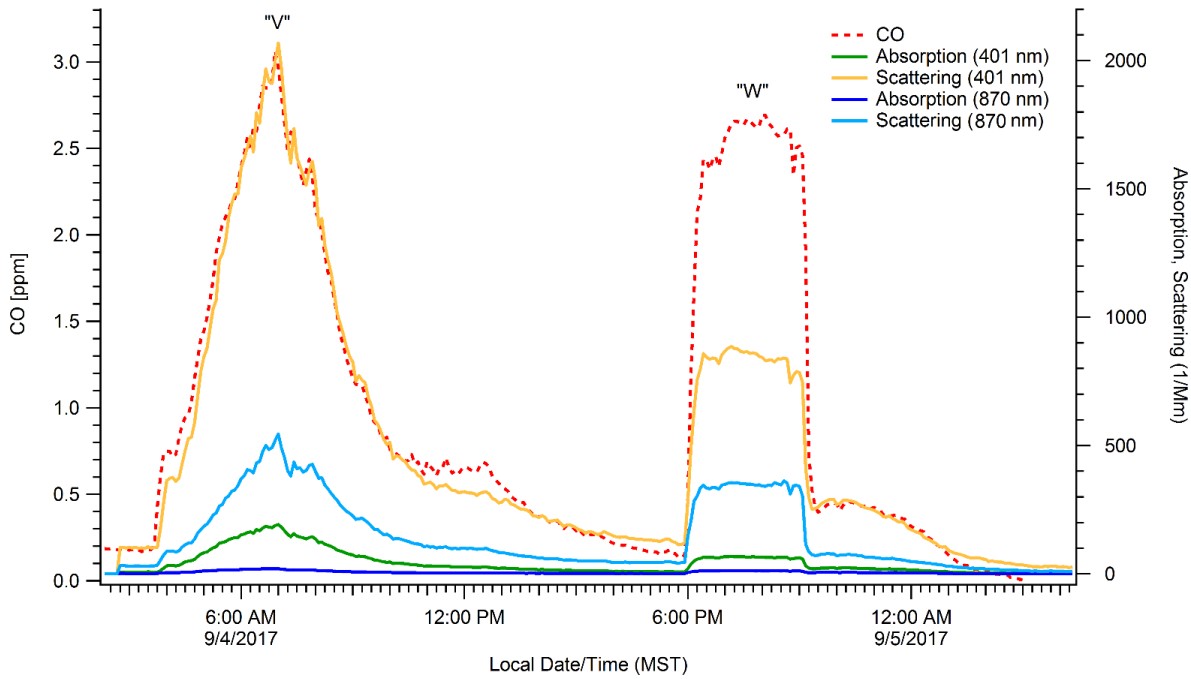

Figure 6. High resolution (5-minute) time series of smoke-impacts measured in Missoula over Labor Day weekend.

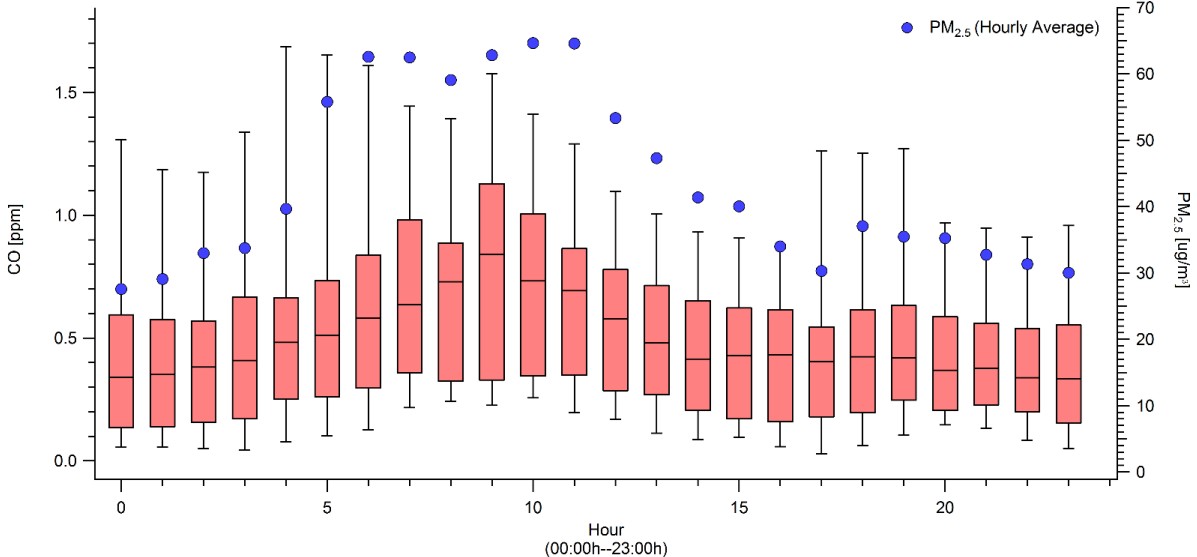

Figure 7. Diurnal plot of CO and PM$_{2.5}$.





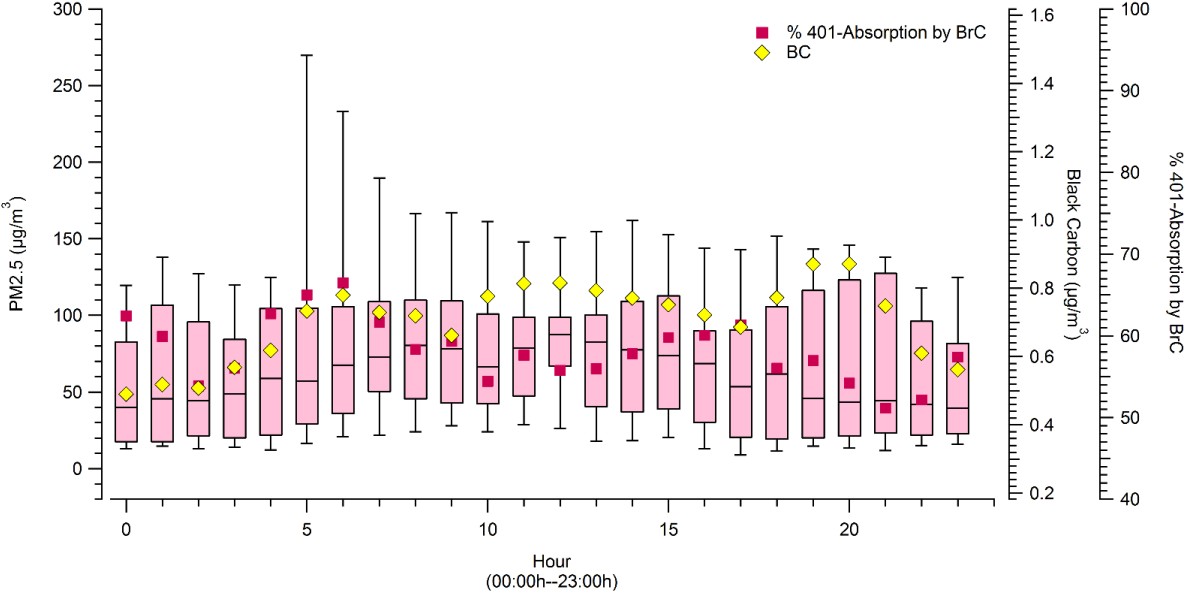

Figure 8. Diurnal plot of average PM$_{2.5}$, hourly average % 401-Absorption by BrC, and hourly average BC.

