# Peer review of "In-situ measurements of trace gases, PM, and aerosol optical properties during the 2017 NW US wildfire smoke event"

_Atmospheric Chemistry and Physics, 2018_

## Referee Comment (RC1) · Anonymous Referee #1 · 20 Dec 2018

GENERAL COMMENTS

This study reports emission ratios and optical properties observed in smoke from wildland fires of various ages. The study measured $\sim$500 hours of smoke impacts, making it very thorough in terms of sampling different ages, a variety of origins, and presumably a diversity of fire behavior. The study methods were appropriate, the presentation and discussion is thorough, yet concise. The authors have been careful to properly qualify their conclusions and have clearly pointed out limitations of the data. The discussion and conclusions are supported by the data presented. I found only one general issue that must be addressed. The comparison with prescribed fires somewhat overstates

the prescribed fire versus wildfire PM differences. The authors have compared their measurements versus all aircraft measured prescribed fire measurements in May et al. (2014), which were mostly SE coastal plain understory burns or California chaparral. Only two of the May et al. (2014) aircraft prescribed fires, the Shaver fire and Turtle fire in Montane forest seem to be similar ecosystems / fuel types to the source fires that impacted Missoula during their study. From a smoke/air quality impact perspective, the wildfire vs. prescribed fire tradeoff issue is largely a matter of forest fires in the western US. The duration, fuel loading, and total emissions involved with western forest fires significantly exceeds that of chaparral and sagebrush systems (see e.g. French et al., 2011 San Diego County fires). Therefore, in the context of a wildfire vs. prescribed smoke/air quality comparison the authors should compare their wildfire results with the Shaver fire and Turtle fire from May et al. (2014), for which the forest PM1/CO = 0.011+/- 0.01, about 40% higher than the 0.08 value presented in Table 5 and used the discussion. Likewise the Shaver / Turtle fire BC/PM1 is 0.006, much closer to that observed in the current study and similar to Liu et al. (2017).

French, N. H. F., et al. (2011), Model comparisons for estimating carbon emissions from North American wildland fire, J. Geophys. Res., 116, G00K05, doi:10.1029/2010JG001469.

SPECIFIC COMMENTS

1. P4, Ln 27-37: Was calibration conducted using the same flow set-up as ambient data, i.e. through scrubber and diffusion drier? Any estimate of particle loss based on other studies?

2. P6, L4-21: Please state the criteria used to define smoke impacted periods of sampling?

3. P6, L4-21: Was a diurnal variation observed in CH4 for background conditions (e.g. due to constant source + varying mixed layer depth)?

4. P7, L33-38 & Table 2: The authors should include CH4/CO ratios form Urbanski (2013) as the wildfires reported on in that study are most similar to the MT/ID/BC fires that impacted Missoula.

5. P7, L 37: "Our higher study average ER of CH4 is indicative of smoldering, or specifically glowing combustion (Yokelson et al., 1997)." This statement implies CH4/CO for glowing combustion is different from smoldering pyrolysis, which is at odds with ground-based field study of Reisen et al. (2018). Please comment on this apparent discrepancy. (Reisen, F., Meyer, C. P., Weston, C. J., & Volkova, L. (2018). JGR - Atmospheres, 123, 8301–8314. https://doi.org/10.1029/2018JD028488 )

6. P9, L1: Reference needed.

7. P9, L14-15: Should note that Hobbs et al. (1996) were mostly prescribed fires of logging slash.

8. How robust is BC = f(MCE) from Selimovic ?

9. Fig 5. Adding date labels to a few ticks on the x-axis would be helpful.\

10. P9, L16-23: Does "annual" refer to 2011 or average over some period of time?

11. Section 3.5. Please note the value of PM2.5/CO over these periods

12. P13, L29-30: This should be restated, prescribed fires do not allow control over dispersion conditions, but allow one to ignite fires when dispersion conditions are favorable and/or manipulate ignition in a manner that enhances dispersion, e.g. mass ignition that puts smoke above mixed layer.

TECHNICAL

Mixing of units notation, superscripts and "/", e.g. L min^-1 and L/min, throughout paper

P4, L36: missing "nm" after 401

P5, L13: "BrC" should not be subscript

P10, L35: missing "nm" after 401

---

## Referee Comment (RC2) · Anonymous Referee #2 · 20 Dec 2018

This manuscript presents measurements of some aerosol properties and some trace gases in Missoula (US) during approx. one month in August-September 2017. During this period the measurement location was affected by several smoke plumes from wild fires. Some of the fire locations were identified, but several plumes represent aged regional smoke containing emissions from various sources. Altogether this data set contains approx. 500 h of in-plume measurements and can provide valuable information on statistics of flaming vs. smoldering combustion on regional scale. However, the methods need to be described in more detail and different sources of uncertainty have to be assessed before this manuscript can be accepted in ACP.

[Figure]
Major comments

My main concern is that uncertainties in the analysis are not well quantified. Uncertainties for individual instruments are presented in Section 2, but uncertainty estimates are not presented for any of the data points in the graphs.

Furthermore, it is not clear how "smoke-impacted" periods are distinguished from non-smoke periods. For instance for peak G in Fig. 1: the "smoke-impacted" BC and CO concentrations during afternoon hours are lower than during the following "non-smoke period". Reliable differentiation between "smoke-impacted" and background periods is essential for accurate definition of excess concentrations and excess mixing ratios especially for more diluted regional smoke (e.g. peaks M, N, R, T in Fig. 1).

Many of the "smoke-impacted" periods last 24h or more. In such cases any diurnal variability in background CO, BC and PM2.5 will be a source of uncertainty, as background is apparently estimated with linear interpolation (see page 6, line 9). Can you estimate how large is the uncertainty in excess mixing ratios due to assumed linear change in background during long smoke-impacted periods?

One more source of uncertainty, which is not very well constrained, is the effect of 3.2km distance between PM2.5 measurements and other measurements. At 1h resolution and for regional scale smoke the distance is probably not an issue, but for the relatively fresh plumes (1-2 h) that distance can make a difference. Is there any difference in the correlation between scattering and PM2.5 for diluted and fresh plumes?

It seems that at the moment only one integrated excess mixing ratio is defined for each smoke-impacted period (page 6, line 9-11). However, many of the smoke-impacted periods represent considerable temporal variability. I recommend calculating excess mixing ratio at e.g. 1h or 5min temporal resolution, which would allow presenting also standard deviation (or other measure of in-plume variability) in addition to mean values in Supplementary Table 1. I think this approach would give also more representative study-average statistics. With the current approach short smoke-impacted periods

have equal weight to long periods in the study average.

Please include also scattering/CO ratio in the analysis. I believe this would be a valuable reference in the future.

Minor comments

Please indicate the units for excess mixing ratios. Are mass concentrations given in prevailing conditions or e.g. STP?

Page 5, line 4. It seems that no truncation error correction was applied to the scattering coefficient. Please discuss shortly the uncertainty in SSA.

Page 5, line 8. Please define SSA based on scattering and absorption coefficients (Babs, Bscat defined on page 4, line 12).

Page 6, line 20-21: "Other approximate metrics of the relative amount of flaming to smoldering such as BC/CO or CH4/CO can still be used". Are these ratios calculated as excess mixing ratio or plain concentration ratio? Please make sure that excess concentrations are always indicated with a delta (also in Figures) - now it seems that most excess mixing ratios are written without delta, i.e. as plain concentration ratio.

Page 8, line 3 and Fig. 2. Are there any previous studies to compare CH4/CO vs. BC/CO dependency?

Page 9, line 9. I agree, but the relationship between MCE and BC/CO is not linear (e.g. Vakkari et al., 2018). Can you estimate the MCE range from BC/CO in your case?

Page 9, line 15. "The Selimovic et al. lab average" Year missing in reference, please check.

Page 9, line 24-25. "Changes in the PM/CO ratio as a plume ages can be used as a metric for the net effect of secondary formation or evaporation of organic and inorganic aerosol (Yokelson et al., 2009; Akagi et al., 2012; Jolleys et al., 2012; Vakkari et al., 2014)." This method was recently applied by Vakkari et al. (2018) as well; you may

consider adding a reference.

Page 9, line 28. "Further our lower BC/CO ratio suggests enhanced smoldering, which should increase the PM/CO." The observations by Vakkari et al. (2014, 2018) seem to indicate the opposite: fresh emission PM/CO decreasing with increasing smoldering. PM emission factor does increase with increasing smoldering, though.

Page 10, line 2-3. "The BC/PM ratio also allows for an estimate of ambient BC from ambient PM data when BC isn't measured, but caution is needed since PM may not be conserved as long as BC." BC fraction may also depend on combustion characteristics (c.f. Vakkari et al., 2014).

Page 10, line7-8. "A previous study found that smoldering combustion emits anywhere between 4-49 times more PM than flaming combustion (Kim et al., 2018)," It seems that Kim et al. (2018) measured total PM (no size cut in inlet), which could be pointed out here. I would expect PM2.5 or PM1 emission variability be a bit less than TSP.

Page 12, line 12-13. "Figure 5 shows a moderate increasing trend in the SSA at 870 nm, but no significant trend in the SSA at 401 nm." Please state how you checked for statistically significant trend.

Page 12, line 29. "smoke was mostly sourced from a local fire (Rice Ridge)." How far was the fire? Can you estimate the smoke age?

Page 12, line 29. "Our peak-integrated proxy for particle size (4.02, smaller particle size)" Please describe the "peak-integrated proxy for particle size" in Section 2.

Figure 6 (case study). Please add a second panel with high-resolution excess mixing ratios (BC/CO, PM2.5/CO, scattering/CO, trace gases/CO) so that the reader can compare the two peaks.

Page 13, Section 3.6 Diurnal Cycles. I would expect diurnal cycle to be important for near-fire measurements due to diurnal variation in the emissions (e.g. Saide et al., 2015), oxidation and dilution. However, I would not expect much difference in aged

regional smoke, whether it is observed during morning or evening hours. Here, focusing on extensive properties (PM2.5, BC, CO) is problematic as they depend mostly on dilution. I wonder if the diurnal cycle in Figure 7 has a small increase in morning only because more fresh plumes happened to reach the measurement site during morning hours. I recommend removing this section or concentrating on fresh plumes (e.g. CO > 0.5 or 1 ppm) and intensive properties (excess mixing ratios).

Page 14, line 11-13. "Despite our lower BC/CO ratio our PM/CO ratio was about half that measured in fresh smoke from aircraft. This suggests that OA evaporation, at least near the surface, may typically reduce PM air quality impacts on the time scale of several days." I do not think you can draw such a straightforward conclusion, as PM/CO ratio decreases with decreasing BC/CO. If both fuel and BC/CO are equal, then a lower PM/CO in aged smoke would suggest primary aerosol evaporation. Please check also abstract (page 1, line 18-22).

It seems that all linear fits are calculated with ordinary least squares method, which assumes that there is no uncertainty in x-direction. At least for Figs. 2, 3 and S1 a bivariate method would be more appropriate (see e.g. Cantrell et al., 2008).

Please combine Tables 1 and 5 to avoid repetition. Please also check that you have defined the values in parenthesis in all Table captions. Is the study average a mean of enhancement ratios defined for each plume?

References

Cantrell, C. A.: Technical Note: Review of methods for linear least-squares fitting of data and application to atmospheric chemistry problems, Atmos. Chem. Phys., 8(17), 5477–5487, doi:10.5194/acp-8-5477-2008, 2008.

Saide, P. E., Peterson, D. A., da Silva, A., Anderson, B., Ziemba, L. D., Diskin, G., Sachse, G., Hair, J., Butler, C., Fenn, M., Jimenez, J. L., Campuzano-Jost, P., Perring, A. E., Schwarz, J. P., Markovic, M. Z., Russell, P., Redemann, J., Shinozuka, Y., Streets, D. G., Yan, F., Dibb, J., Yokelson, R., Toon, O. B., Hyer, E. and Carmichael, G. R.: Revealing important nocturnal and day-to-day variations in fire smoke emissions through a multiplatform inversion, Geophysical Research Letters, 42(9), 2015GL063737, doi:10.1002/2015GL063737, 2015.

Vakkari, V., Beukes, J. P., Dal Maso, M., Aurela, M., Josipovic, M. and van Zyl, P. G.: Major secondary aerosol formation in southern African open biomass burning plumes, Nature Geosci., 11, 580–583, doi:10.1038/s41561-018-0170-0, 2018.
* * *

---

## Referee Comment (RC3) · Anonymous Referee #3 · 31 Dec 2018

This manuscript presents a major wildfire aged smoke measurement of some aerosol properties and trace gases in Missoula (US) during August-September 2017. During this period the measurement location was affected by several smoke plumes from wild fires, more importantly a smoldering and nighttime fire chemistry case is presented. Model back trajectories and satellite retrievals allowed for some of the fire locations to be identified and investigated. In summary, this data set presented here contains approx. 500 h of ground-based plume measurements and can provide valuable infor-

mation on statistics for modeling and emission factors based on flaming vs. smoldering combustion on a regional scale. The prescribed burning comparisons are an interesting start to a much-needed solution. I think this paper is acceptable but could benefit from a deeper look into the implications for modeling use via smoldering and nighttime chemistry.

Major comments

Page 3 line 15: The author indicates that this can be used to inform model mechanisms; however, outside of presenting numbers for ratios (which can and is helpful) without context of in what way to use these ratios. Meaning, all numbers are not created equal, in what modeling scenario should these new numbers or measurements be applicable? Are these numbers for nighttime generated smoke? Can one use these numbers when a fire is detected at night or during the day and expected to be smoldering? E.g. page 6 line 5: "time series of mixing ratios" is helpful to point out in detail. E.g. BC/CO as a function of distance would be helpful.

Page 4 line 3-5; brief discussion of the uncertainties; there needs to be more in this paper about those uncertainties associated with each calculation and its use in a modeling platform or intended use.

Page 6, line 18-21 MCE is not a good indicator of flaming vs smoldering compared to BC and CH4 ratios to CO, needs a citation, unless you are planning on providing evidence in this paper of this using the data collected?

Page 7, line 18-27 it seems that the authors had an opportunity with this data set to take a look into the various composition of fuels and impacts on transported chemistry. The small caveat to this is that hysplit will not likely give you 100% certainty on the origin, but with the fires that were identified, I would have liked to see an attempt to separate out measured emissions vs fuel types. This could potentially be a nice case study for Lolo Peak fire and Rice Ridge fire. As this fuels composition could be one explanation of the presented results differences between the other studies.

Page 8, line 17 "time since emission" I would have like a deeper dig into this as the results all hinge upon the accuracy of this. The authors claim the smoke came from late afternoon to nighttime but do not show this anywhere outside of the supplemental materials. And since hysplit does not include full chemistry it seems odd to use it to look at full chemistry transported, but as you indicated the ratios compared to the relatively conserved CO should be okay.

Page 8, line 35 the separation of smoldering vs flaming vs residual smoldering is difficult, particularly in modeling and source attribution. If there was a ratio or tracer method that was found to actually indicate one of the other this was not clear to me reading this. It appears the distinction was made based off time of day (and one case presented grew at night), knowledge of fires state, and measured chemistry. Which is nice but going forward most cases wont have all that information.

Page 9, line 17. It appears that this study used only three heights to initialize hysplit, but did not indicate why those heights where chosen (if it was based purely on the elevation of the terrain then that makes sense). However, it does not include the effects of plume rise? As smoldering smoke tends to pool near the surface but can reach higher elevations, and vice versa for flaming smoke.

Consider the references

Wilkins JL, Pouliot G, Foley K, Appel W, Pierce T (2018) The impact of US wildland fires on ozone and particulate matter: a comparison of measurements and CMAQ model predictions from 2008 to 2012. International Journal of Wildland Fire, https://doi.org/10.1071/WF18053.

Zhou L, Baker KR, Napelenok SL, Pouliot G, Elleman R, O'Neill SM, Urbanski SP, Wong DC (2018) Modeling crop residue burning experiments to evaluate smoke emissions and plume transport. Science of the Total Environment 627, 523-533, https://doi.org/10.1016/j.scitotenv.2018.01.237.

Page 9, line 33 aging and/or higher average temperatures at lower elevation may encourage some OA evaporation and reduce downwind PM impacts. This line is very interesting and should be expanded upon, as it's a critical finding from this study. What here is indicated as higher average temperatures? Is this flaming stage or just hot temperatures in the atmosphere as the plume ages? (page 10, line 12-15 also are confusing for the same reason "and thus strongly cooling"). Furthermore, can a statement be made in this section about smoldering plumes traveling in hotter temperatures or temperature of plume on evaporation of PM? This point would be good to attempt to relate to prescribed burns, as the emissions tend to be more toxic (or higher for PM) from the incomplete combustion and lower temperatures of burns and therefore longer smoldering time periods.

Also, for the section 3.2 (page 10, line 3-5) are the authors discussing BC on average or BC for smoldering cases. It seems from the way its written that this ratio is for smoldering and the one presented in Liu et al. is for flaming? Could there be a statement made such as BC/PM < x is expected to be from smoldering while BC/PM > x is expected to be flaming?

Page 13, line 20 It states that a possibly explanation is that more BC is being generated during the day, however it transported to the site over night inorder to arrive by 5am. Or is this statement meant to mean, the transported plume that remained over Missoula cooked during the daytime hours and generated more BC during the daytime while at Missoula?

Minor comments

There is a need for a careful defining of terms. Some terms are used before they are defined, and others are never defined. And I believe all terms should be defined that are used in the abstract. E.g. BrC is used on page 1 line 23 and defined later on line 28; "US" is used on page 1 line 37 and is not defined. The authors need to decide whether or not to abbreviate which terms and remain consistent, e.g. Biomass burning

appears as BB sometimes and other times not, also Air quality is sometimes AQ.

A through grammar check is needed. There are some run on sentences and some missed placed commas and periods. E.g. page 2 line 3-10 very long run-ons.

Page 10, line 35 does this ratio come with a trend or can expect numbers be inferred?

Page 11, line 36 what is meant by "870 nm is unity to a good approximation " the transitions at the end of paragraphs in my opinion are not needed (e.g. Page 13, line 12) " which we examine next"

---

## Author Comment (AC1) · 25 Feb 2019

Response to Referee #1

We thank the Referee for all their comments, which have helped improve the paper as described below. The Referee suggestions are shown in full along with our detailed response/revisions in an "R#, A#" format next.

**R1.** GENERAL COMMENTS: This study reports emission ratios and optical properties observed in smoke from wildland fires of various ages. The study measured ~500 hours of smoke impacts, making it very thorough in terms of sampling different ages, a variety of origins, and presumably a diversity of fire behavior. The study methods were appropriate, the presentation and discussion is thorough, yet concise. The authors have been careful to properly qualify their conclusions and have clearly pointed out limitations of the data. The discussion and conclusions are supported by the data presented. I found only one general issue that must be addressed. The comparison with prescribed fires somewhat overstates the prescribed fire versus wildfire PM differences. The authors have compared their measurements versus all aircraft measured prescribed fire measurements in May et al. (2014), which were mostly SE coastal plain understory burns or California chaparral. Only two of the May et al. (2014) aircraft prescribed fires, the Shaver fire and Turtle fire in Montane forest seem to be similar ecosystems / fuel types to the source fires that impacted Missoula during their study. From a smoke/air quality impact perspective, the wildfire vs. prescribed fire tradeoff issue is largely a matter of forest fires in the western US. The duration, fuel loading, and total emissions involved with western forest fires significantly exceeds that of chaparral and sagebrush systems (see e.g. French et al., 2011 San Diego County fires). Therefore, in the context of a wildfire vs. prescribed smoke/air quality comparison the authors should compare their wildfire results with the Shaver fire and Turtle fire from May et al. (2014), for which the forest PM1/CO = 0.011+/- 0.01, about 40% higher than the 0.08 value presented in Table 5 and used the discussion. Likewise the Shaver / Turtle fire BC/PM1 is 0.006, much closer to that observed in the current study and similar to Liu et al. (2017).

French, N. H. F., et al. (2011), Model comparisons for estimating carbon emissions from North American wildland fire, J. Geophys. Res., 116, G00K05, doi:10.1029/2010JG001469.

**A1.** This is a good comment though we assume the Referee meant a PM/CO of 0.11 and a BC/CO of 0.006. The Turtle and Shaver Fires comprise a small data set and it is not clear that brush, chaparral, or grass fires did not impact us in Missoula. E.g. The Rim Fire burned coniferous forests in the Sierras, but also oak and chaparral. In addition, the SE-US prescribed fires in May et al (2014) were also in coniferous ecosystems and help make a larger prescribed fire data set. The Turtle and Shaver prescribed fires may have burned more understory and less overstory than wildfires in the same ecosystem. We also have to be careful because the prescribed fire frequency (time since last burn) could impact emissions and, most of all, we are comparing fresh prescribed fire smoke (May et al, 2014) to more aged wildfire smoke (Missoula 2017) and prescribed fire emissions typically occur at lower air temperatures. In fact, May et al., (2015) observed a significant decrease of OA/CO after five hours of aging for one of the prescribed fire smoke plumes in their 2014 paper. However, we agree that it is worth acknowledging that the geographic location may combine with vegetation type to influence EF so we have added a discussion of the impact of comparing wildfires to a smaller, potentially more relevant, subset of prescribed fires as suggested by the referee.

Changes:
We added a column to Table 5 that shows the data averaged over just the Turtle and Shaver fires.

P 13, L36 Old text: The available PM/CO data for wildfires is consistently higher than for prescribed fires, which has air quality and land management implications.

The available PM/BC ratios are consistently ~20 times higher for wildfires, than prescribed burns, confirming that wildfire smoke is overwhelmingly more organic, which is important partly because many optical properties scale with the BC/OA ratio. In general, our ground-based wildfire study confirms the earlier airborne indications that prescribed fires are less smoky but also less cooling than wildfires.

New text: The $\Delta PM/\Delta CO$ values for fresh wildfire smoke in Liu et al. (2017) and aged wildfire smoke (this study) are about three and 1.5 times higher than $\Delta PM/\Delta CO$ for fresh smoke from prescribed fires in May et al. (2014) when comparing to all their US prescribed fires (Tab. 5). For only prescribed fires in western US mountain coniferous ecosystems (last column Tab. 5), the $\Delta PM/\Delta CO$ for fresh smoke is close to our value for aged wildfire smoke. However, May et al. (2015) noted that $\Delta PM/\Delta CO$ decreased by about a factor of two after several hours of aging on at least one prescribed fire.

The $\Delta BC/\Delta CO$ for prescribed fires is higher than the wildfire average by a factor of ~9 (all prescribed fires) or ~4 (last column), roughly suggesting a higher MCE for prescribed fires. Ignoring smoke age, the $\Delta BC/\Delta PM$ for prescribed fires is higher than the wildfire average by a factor of ~20 (all prescribed fires) or ~6 (last column). The $\Delta BC/\Delta PM$ observations suggest that wildfire smoke is overwhelmingly more organic, which is important partly because many optical properties scale with the BC/OA ratio (Saleh et al., 2014). In general, our ground-based wildfire study confirms the earlier airborne indications that prescribed fires are less smoky but also less cooling than wildfires. Differences in smoke production and chemistry between wild and prescribed fires should be researched more and have air quality and land management implications.

Reference
May, A. A., Lee, T., McMeeking, G. R., Akagi, S., Sullivan, A. P., Urbanski, S., Yokelson, R. J., and Kreidenweis, S. M.: Observations and analysis of organic aerosol evolution in some prescribed fire smoke plumes, Atmos. Chem. Phys., 15, 6323-6335, doi:10.5194/acp-15-6323-2015, 2015.

SPECIFIC COMMENTS

**R2.** P4, Ln 27-37: Was calibration conducted using the same flow set-up as ambient data, i.e. through scrubber and diffusion drier? Any estimate of particle loss based on other studies?

**A2.** All the calibrations were done with the same sample line, cyclones, drier, and scrubber in the same location. We only report data where instrument pressures, flow rates, leak checks, etc passed QC checks, and for 401 nm, data that was collected between calibrations where the AAE was within +/- ~2-3 percent of one for fresh propane torch soot. We did not have the specialized equipment to measure any particle losses in the diffusion drier, but using the same drier and scrubber in 2018 with a PM2.5 cyclone gave mass-scattering coefficients for 2.5 micron

scattering divided by 2.5 mass that were very close to the middle of the range of numerous other studies indicating that minimal losses are occurring in the drier and scrubber. However, this was an important comment. At the time we set up the PAXs we were aware of websites (at least 3) that suggested drier losses were "minimal." However, upon re-investigating, only two of the three websites still make this claim and a recent paper briefly includes a somewhat relevant measured size-independent particle transmission efficiency (Miyakawa et al., 2017) for their diffusion drier of 84 +/- 5%. We have not applied a correction to our data because we did not measure anything specific to our setup. Referee #2 also brought up one more source of uncertainty; truncation error in the nephelometer. We added new text at P5, L25 to address several poorly characterized sources of error together after defining the relevant parameters.

New text:

P4. L30-32 text changed to: "The scrubber and drier were refreshed before any signs of deterioration were observed (e.g. color change). The diffusion based designs will cause small particle losses, but losses were not explicitly measured."

P5, L24: A few other sources of uncertainty in the measurements and/or calculations are poorly characterized; MAC increases due to coatings, potential particle losses in the drier or scrubber, and truncation error in the nephelometer. Mie calculations provided by the manufacturer suggest the scattering could be underestimated by about 1% at 870 nm and 2.5% at 401 nm due to truncation error (J. Walker, private communication). This would reduce the mass scattering coefficients (Sect. 3.4) and typically, a 1% reduction in scattering would imply approximately a tenth of a percent of value underestimate of SSA. Miyakawa et al. (2017) reported a size-independent particle transmission up to 400 nm of 84±5% in their diffusion drier. Larger particles may be transmitted more efficiently. We did not measure size distribution or transmission efficiency in this study and thus, we did not adjust the data. Size-independent particle losses would reduce scattering, absorption, and derived BC, but should have only a small impact on SSA or AAE. Unlike particle losses, an increased MAC due to "lensing" via coatings would inflate BC values by up to ~30% (Pokhrel et al., 2017).

Reference: Miyakawa, T., Oshima, N., Taketani, F., Komazaki, Y., Yoshino, A., Takami, A., Kondo, Y., and Kanaya, Y.: Alteration of the size distributions and mixing states of black carbon through transport in the boundary layer in east Asia, Atmos. Chem. Phys., 17, 5851-5864, https://doi.org/10.5194/acp-17-5851-2017, 2017.

**R3.** P6, L4-21: Please state the criteria used to define smoke impacted periods of sampling?

**A3.** Any sustained period with $PM_{2.5}$ well above 12.5 µg/m$^3$, which EPA defines as the upper limit for good air quality, was included in one of the smoke events. Referee #2 commented on the divisions between events. We did not apply a formal algorithm. Instead, for instance, when high PM levels decreased to a local minimum, or more sustained values, near or below the "good" air quality level (12.5 µg/m3) we took this as the end of the "event." In some cases a post-event "cleaner period" was sustained, but sometimes a single point is the end of one event and the start of another. We also elected not to integrate some small or brief peaks that sometimes occurred after adjacent larger peaks. For instance, a small peak after peak G, was not included because of low S:N. The last peak was integrated up to where the CO measurement

failed. We verified several times that the integrals for events are dominated by the large values and insensitive to small shifts in the endpoints at lower levels.

P7, L20: Added: Sustained periods when $PM_{2.5}$ was elevated well above the 12.5 µg/m$^3$ EPA standard for "good" air quality were designed as events and assigned a letter in Fig. 1 and Tab. S1.

**R4.** P6, L4-21: Was a diurnal variation observed in CH4 for background conditions (e.g. due to constant source + varying mixed layer depth)?

**A4.** From 2017 and 2018 there is some variability in $CH_4$ during smoke-free periods, but it is not well-defined enough to confidently calculate a new baseline under peaks and we have no evidence that it contributes overall bias to the integrals. Most likely the variability contributes to a higher standard deviation for our measured ΔCH4/ΔCO ratios than we might have seen otherwise. This topic is also addressed in some detail in the response to Referee #2. Basically, the concept of a measureable background was usually not applicable due to the widespread (often synoptic scale) impacts.

**R5.** P7, L33-38 & Table 2: The authors should include CH4/CO ratios form Urbanski (2013) as the wildfires reported on in that study are most similar to the MT/ID/BC fires that impacted Missoula.

**A5.** We added the Urbanski CH4/CO (0.0946 +/- 0.0108) to Tab.1. Note: we were also impacted by fires in WA, OR, CA.

**R6.** P7, L 37: "Our higher study average ER of CH4 is indicative of smoldering, or specifically glowing combustion (Yokelson et al., 1997)." This statement implies CH4/CO for glowing combustion is different from smoldering pyrolysis, which is at odds with ground-based field study of Reisen et al. (2018). Please comment on this apparent discrepancy. (Reisen, F., Meyer, C. P., Weston, C. J., & Volkova, L. (2018). JGR - Atmospheres, 123, 8301–8314. https://doi.org/10.1029/2018JD028488 )

**A6.** Reisen et al "visually" "broadly" sorted samples into glowing or pyrolysis, but pure pyrolysis cannot actually occur alone since pyrolysis requires heat. Glowing can occur "alone" briefly in a lab if no fresh fuels are left to pyrolyze. Yokelson et al., (1997) reported that CH$_4$ was enhanced from glowing compared to other organic gases, but CO could have also been enhanced from glowing so the Referee is correct that additional analysis would be needed to scope out effects on the ☐CH$_4$/☐CO ratio due to glowing/pyrolysis. That is beyond scope of this study, especially since fuel type may have an impact and authentic field conditions make it harder to isolate processes than in lab. We decided to remove any speculation about smoldering sub-types since the sub-type of smoldering is secondary here to our main point that ☐CH$_4$/ ☐CO tends to increase with smoldering. We do note that ☐CH$_4$/☐CO increased with smoldering (lower MCE) in Reisen et al (as in many other studies), which supports our interpretation of our Figure 2.

Change in text: We deleted ", or specifically glowing combustion" and added the Reisen et al reference on line 38.

**R7.** P9, L1: Reference needed.

**A7.** The references on line 2 supported both of the first two sentences. We moved Bond 2004 reference to the previous line to clarify that literature support exists for both sentences.

**R8.** P9, L14-15: Should note that Hobbs et al. (1996) were mostly prescribed fires of logging slash.

**A8.** We compared specifically to the subset of fires described a wildfires as now noted:

Old text: "The Hobbs et al. (1996) is notably"

New text: "The Hobbs et al. (1996) average value for their two fires specifically identified as wildfires is notably"

**R9.** How robust is BC = f(MCE) from Selimovic ?

**A9.** BC/CO correlates with MCE, but with considerable noise and in non-linear fashion. To acknowledge this, on page 9, line 16 we appended ", which tends to enhance BC emissions." We added a plot of BC/CO versus MCE from the Selimovic study to Fig. 2 in response to this comment and Referee #2 and some brief new text described in that response

**R10.** Fig 5. Adding date labels to a few ticks on the x-axis would be helpful.

**A10.** Done.

**R11.** P9, L16-23: Does "annual" refer to 2011 or average over some period of time?

**A11.** We changed "Liu et al. (2017) calculated an annual CO production from western US wildfires of $5240 \pm 2240$ Gg, which they reported was in good agreement with an EPA estimate from the 2011 National Emissions Inventory (4894)."

to "Liu et al. (2017) calculated an average annual CO production from western US wildfires for 2011-2015 of $5240 \pm 2240$ Gg, which they reported was in good agreement with an EPA estimate based on a similar burned area in the 2011 National Emissions Inventory (4894)."

On line 23 we added the year (2006) for the Mao et al (2015) study.

**R12.** Section 3.5. Please note the value of PM2.5/CO over these periods.

**A12.** Unfortunately, the PM monitor had its few missing hourly values during peak "W".

**R13.** P13, L29-30: This should be restated, prescribed fires do not allow control over dispersion conditions, but allow one to ignite fires when dispersion conditions are favorable and/or manipulate ignition in a manner that enhances dispersion, e.g. mass ignition that puts smoke above mixed layer.

**A13.** This sentence now ends "… and can be ignited when conditions are favorable for minimizing air quality impacts (Liu et al., 2017)"

TECHNICAL

**R14.** Mixing of units notation, superscripts and "/", e.g. L min^-1 and L/min, throughout paper

**A14.** Fixed

**R15.** P4, L36: missing "nm" after 401

**A15.** Fixed

**R16.** P5, L13: "BrC" should not be subscript

**A16.** This could be OK or could be parenthetical format, but be consistent.

**R17.** P10, L35: missing "nm" after 401

**A17.** Fixed

**Voluntary updates.**

We made three additional voluntary minor changes (described next). We did some very light editing (updating references, added a missing word, etc) that is indicated in "track changes."

P2, L29: Added Tomaz et al., 2018 reference.

P6, L37: Changed "British Columbia experienced a record fire season…" to "Over 1.2 million ha burned in British Columbia in 2017." The previous record was broken in 2018.

P12, L24, after Ansmann citation: We re-inserted a link that had been accidentally removed to a NASA website that described how Labor Day weekend smoke from the NW US reached Europe.

Table 4. The MSC and MAC values between 870 and 401 nm were adjusted slightly using a more accurate method of extrapolation. We note that both calculation methods produce MSCs in excellent agreement with the literature when used with our 2018 data that was collected with PM2.5 cyclones on the PAXs. With both procedures, the MSC values are lower with PM1 cyclones on the PAXs. We think the PM1 cyclones likely do a good job of isolating the combustion generated aerosol, but that super-micron dust and vegetative debris gets entrained in smoke plumes, transported, and affects the optical properties, which has prompted us to switch to PM2.5 cyclones for continued monitoring. Change "linear" in text to indicate based on power law fit.

P11, L35: Old text: "…were calculated using a linear regression using the calculated averages."

New text: "…were calculated with a power law fit using the calculated average."

Figure 3. Added caption: "b) Lab averaged BC/CO ratio versus modified combustion efficiency (MCE) separated into bins by 0.1 of MCE.

Figure 4. We had the caption for parts a and b reversed and that has been corrected.

Figure 7. Added "shown for the entirety of the monitoring period" to caption.

Figure 8. Added "BC and PM shown for the entirety of the monitoring period, but %401-Absorption by BrC only shown for when the PAX 401 was operational."

---

## Author Comment (AC2) · 25 Feb 2019

Response to Referee #2

We thank the Referee for all their comments, which have helped improve the paper as described below. The Referee suggestions are shown in full along with our detailed response/revisions in an "R#, A#" format next.

**R1.** This manuscript presents measurements of some aerosol properties and some trace gases in Missoula (US) during approx. one month in August-September 2017. During this period the measurement location was affected by several smoke plumes from wild fires. Some of the fire locations were identified, but several plumes represent aged regional smoke containing emissions from various sources. Altogether this data set contains approx. 500 h of in-plume measurements and can provide valuable information on statistics of flaming vs. smoldering combustion on regional scale. However, the methods need to be described in more detail and different sources of uncertainty have to be assessed before this manuscript can be accepted in ACP.

**A1.** We appreciate the positive feedback and also briefly note that even periods dominated by individual fires were not "pure" and affected by some mixing of sources.

Major comments

**R2.** My main concern is that uncertainties in the analysis are not well quantified. Uncertainties for individual instruments are presented in Section 2, but uncertainty estimates are not presented for any of the data points in the graphs.

**A2.** We added representative error bars in the figures.

**R3.** Furthermore, it is not clear how "smoke-impacted" periods are distinguished from non-smoke periods. For instance for peak G in Fig. 1: the "smoke-impacted" BC and CO concentrations during afternoon hours are lower than during the following "non-smoke period". Reliable differentiation between "smoke-impacted" and background periods is essential for accurate definition of excess concentrations and excess mixing ratios especially for more diluted regional smoke (e.g. peaks M, N, R, T in Fig. 1).

**A3.** We did not apply a formal algorithm. Instead, for instance, when high PM levels decreased to a local minimum, or more sustained values, near or below the "good" air quality level (12.5 µg/m3) we took this as the end of the "event." In some cases a post-event "cleaner period" was sustained, but sometimes a single point is the end of one event and the start of another. We also elected not to integrate some small or brief peaks that sometimes occurred after adjacent larger peaks. For instance, a small peak after peak G, was not included because of low S:N. The last peak was integrated up to where the CO measurement failed. We verified several times that the integrals for events are dominated by the large values and insensitive to small shifts in the endpoints at lower levels.

P7, L25, new text: ~Sustained periods when PM2.5 was elevated well above 12.5 were designated as events and assigned a letter in Fig. 1 and Tab. S1.

**R4.** Many of the "smoke-impacted" periods last 24h or more. In such cases any diurnal variability in background CO, BC and PM2.5 will be a source of uncertainty, as background is apparently estimated with linear interpolation (see page 6, line 9). Can you estimate how large is

the uncertainty in excess mixing ratios due to assumed linear change in background during long smoke-impacted periods?

**A4.** We can only probe the variability in the smoke-free backgrounds by examination of the smoke-free periods in 2017 and now 2018. $CO_2$ doesn't have a repeating pattern and varies substantially so we don't attempt $CO_2$ integrals. $CH_4$ varies enough to add noise to the $\Delta CH_4/\Delta CO$ ratios, which is likely reflected in the large stdev, but not in a systematic way that we can use to justify a non-linear baseline assumption and not in a way that suggests systematic bias from a linear assumption. BC, PM, and CO "bottom out" at levels close to zero during smoke free periods with no evidence of significant background variability. Since we are in the midst of widespread impacts and not adjacent to distinct, "spatially small" plumes, there is no actual instantaneous background that could be measured by comparing inside and outside the smoke. Typically the nearest clean air was hundreds of miles away and probably not a valid background for our site. Using the linear assumption to generate a "calculated background" for estimated excess mixing ratios is standard practice in peak integration and the most complex assumption that we can justify. At the real-time level any single, point excess mixing ratio might have a substantial uncertainty especially on the peak edges, but we have no rigorous way to estimate that. Because the smoke concentrations are so much larger than background (except for methane), it's likely that the error in the peak integrated values are very small.

**R5.** One more source of uncertainty, which is not very well constrained, is the effect of 3.2km distance between PM2.5 measurements and other measurements. At 1h resolution and for regional scale smoke the distance is probably not an issue, but for the relatively fresh plumes (1-2 h) that distance can make a difference. Is there any difference in the correlation between scattering and PM2.5 for diluted and fresh plumes?

**A5.** Even the freshest smoke was spread over wide areas and the concept of a well-defined plume, which we contributed to by using the word "plume" incorrectly is misleading here (vide infra). Both the BC vs PM and the supplemental MSC plots indicate good mixing across most of the concentration range, but with some increased scatter for higher values that could be due to concentrated pockets embedded in "smoke fronts" that arrived at the separated measurement sites at offset times. However, there are not enough of these high points to warrant a separate analysis nor do they provide evidence of bias from using the whole data set. The r-squared values are good in all these plots, they provide some idea of the uncertainty in the ratio, and we also added the uncertainty in the slopes. We've checked the text and tried to use the word "plume" more carefully.

The following text was changed:

P1, L29; P9, L33; P12, L10; P14, L17: "plume" to "smoke"

P11, L6: "…aging time for multiple plumes is…" to "…average age of mixed-age smoke…"

**R6.** It seems that at the moment only one integrated excess mixing ratio is defined for each smoke-impacted period (page 6, line 9-11). However, many of the smoke-impacted periods represent considerable temporal variability. I recommend calculating excess mixing ratio at e.g. 1h or 5min temporal resolution, which would allow presenting also standard deviation (or other measure of in-plume variability) in addition to mean values in Supplementary Table 1. I think

this approach would give also more representative study-average statistics. With the current approach short smoke-impacted periods have equal weight to long periods in the study average.

**A6.** We now specify that we used the time-weighted averages of the episode values in the text and in our tables, and note that they are essentially the same as the straight average.

New text:

P7, L28: "Table 1 reports study average ratios weighted by event duration (time-weighted)" old text: "Table 1 reports study average…"

For reasons given above we hesitate to compute real-time excess mixing ratios, but we have added some real-time absolute data to the Labor Day Weekend case study plots in Fig. 6. Again, the smoke levels are so dominant that the ratios between absolute values should be very close to the ratios between excess values.

New Figure 6:

[Figure]

We agree we are curious about the information content at the sub-episode level. However, our site is not in flat terrain impacted by one distinct plume at a time coming with a single wind

direction that allows "hour-resolution" age estimates based on distance to hotspot. In our valley site the flow is often slow to non-existent and highly variable in direction. It's hard to know the relative extent to which transport time is changing during an event. Not only is the horizontal transport complex, but the vertical mixing is complex. For example, inversions are common and mixing smoky free troposphere air down into the boundary layer can't be distinguished from arrival of smoke through the boundary layer a-priori. We can't measure the smoke properties before or after our site. The big picture as far as advancing the interpretation is that we should soon have 3 summers of data to compare to a detailed model and are in discussions with modeling groups to eventually help us with more detailed interpretation as a separate paper.

**R7.** Please include also scattering/CO ratio in the analysis. I believe this would be a valuable reference in the future.

**A7.** We added scattering to CO to Fig. 6.

Minor comments

**R8.** Please indicate the units for excess mixing ratios. Are mass concentrations given in prevailing conditions or e.g. STP?

**A8.** P4, L4 we added "(ppmv)" after "mixing ratio"

P5, L2, before the reference: We added "at ambient temperature and pressure"

P5, L31, after "concentration": We added "$\mu$g m$^{-3}$ at ambient temperature and pressure"

We ensured that units are specified everywhere.

**R9.** Page 5, line 4. It seems that no truncation error correction was applied to the scattering coefficient. Please discuss shortly the uncertainty in SSA.

**A9.** As shown in the reply to Referee #1, the truncation error is believed to be 1-2.5% with about ten times smaller error in the SSA. New text was added to summarize a few error sources:

P5, L24: A few other sources of uncertainty in the measurements and/or calculations are poorly characterized; MAC increases due to coatings, potential particle losses in the drier or scrubber, and truncation error in the nephelometer. Mie calculations provided by the manufacturer suggest the scattering could be underestimated by about 1% at 870 nm and 2.5% at 401 nm due to truncation error (J. Walker, private communication). This would reduce the mass scattering coefficients (Sect. 3.4) and typically. a 1% reduction in scattering would imply approximately a tenth of a percent of value underestimate of SSA. Miyakawa et al. (2017) reported a size-independent particle transmission up to 400 nm of 84±5% in their diffusion drier. Larger particles may be transmitted more efficiently. We did not measure size distribution or transmission efficiency in this study and thus, we did not adjust the data. Size-independent particle losses would reduce scattering, absorption, and derived BC, but should have only a small impact on SSA or AAE. Unlike particle losses, an increased MAC due to "lensing" via coatings would inflate BC values by up to ~30% (Pokhrel et al., 2017).

**R10.** Page 5, line 8. Please define SSA based on scattering and absorption coefficients (Babs, Bscat defined on page 4, line 12).

**A10.** Done.

**R11.** Page 6, line 20-21: "Other approximate metrics of the relative amount of flaming to smoldering such as BC/CO or CH4/CO can still be used". Are these ratios calculated as excess mixing ratio or plain concentration ratio? Please make sure that excess concentrations are always indicated with a delta (also in Figures) - now it seems that most excess mixing ratios are written without delta, i.e. as plain concentration ratio.

**A11.** We've implemented the "Δ" notation consistently throughout the paper text and figures now

**R12.** Page 8, line 3 and Fig. 2. Are there any previous studies to compare CH4/CO vs. BC/CO dependency?

**A12**. Good comment. We think the most valid previous study to compare dependence on MCE to comes from burning western wildfire fuels in the lab where mixing cannot distort MCE (Selimovic et al., 2018). We've added a BC/CO vs MCE plot to Fig. 2. and used it to roughly estimate average MCE for the regional surface level smoke. This topic continues below.

New Fig. 2 plot:

[Figure]

**R13.** Page 9, line 9. I agree, but the relationship between MCE and BC/CO is not linear (e.g. Vakkari et al., 2018). Can you estimate the MCE range from BC/CO in your case?

**A13.** Our BC/CO vs MCE plot is non-linear and qualitatively similar to that in Vakkari et al. It also roughly suggests an MCE below the aircraft value of 0.91.

P9, L17 Old text: "Taken together, this suite of observations is consistent with our ground-based site being impacted by relatively more smoldering combustion compared to the other, mostly airborne, studies."

New text: "Taken together, this suite of observations is roughly consistent with our ground-based site being impacted by relatively more smoldering combustion (MCE ~ 0.87±0.02, based on Fig. 2) than the airborne studies (MCE 0.91 Liu et al., 2017; 0.90 Sahu et al., 2012)."

**R14.** Page 9, line 15. "The Selimovic et al. lab average" Year missing in reference, please check.

**A15.** Done.

**R16.** Page 9, line 24-25. "Changes in the PM/CO ratio as a plume ages can be used as a metric for the net effect of secondary formation or evaporation of organic and inorganic aerosol (Yokelson et al., 2009; Akagi et al., 2012; Jolleys et al., 2012; Vakkari et al., 2014)." This method was recently applied by Vakkari et al. (2018) as well; you may consider adding a reference.

**A16.** We added the suggested reference on P9, L25.

**R17.** Page 9, line 28. "Further our lower BC/CO ratio suggests enhanced smoldering, which should increase the PM/CO." The observations by Vakkari et al. (2014, 2018) seem to indicate the opposite: fresh emission PM/CO decreasing with increasing smoldering. PM emission factor does increase with increasing smoldering, though.

**A17.** This is a valid point. DX/DCO typically increases for smoldering gases (such as CH4) as MCE decreases, but a quick check of the data in several papers shows that PM/CO can increase, stay the same, or even decrease slightly as MCE decreases. We revised the text to indicate that a large "factor of two" drop in PM/CO is not consistent with the known increase in EFPM with MCE.

P9, L28 new text: "Further our lower ☐BC/☐CO ratio su
should preclude a large drop in ☐PM/☐CO (Reisen et al., 2018)."

**R18.** Page 10, line 2-3. "The BC/PM ratio also allows for an estimate of ambient BC from ambient PM data when BC isn't measured, but caution is needed since PM may not be conserved as long as BC." BC fraction may also depend on combustion characteristics (c.f. Vakkari et al., 2014).

**A18.** We changed "an estimate" to "a rough estimate" and (at the end of the sentence) appended "and ☐BC/☐PM is also variable at the source."

**R19.** Page 10, line 7-8. "A previous study found that smoldering combustion emits anywhere between 4-49 times more PM than flaming combustion (Kim et al., 2018)," It seems that Kim et al. (2018) measured total PM (no size cut in inlet), which could be pointed out here. I would expect PM2.5 or PM1 emission variability be a bit less than TSP.

**A19.** The fine mode could vary with MCE more if the super-micron is dominated by entrained dust or vegetative debris. We added more references that make a similar point with fine PM and updated the range to 2-49 in the text.

References

Jen, C. N., Hatch, L. E., Selimovic, V., Yokelson, R. J., Weber, R., Fernandez, A. E., Kreisberg, N. M., Barsanti, K. C., and Goldstein, A. H.: Speciated and total emission factors of particulate organics from burning western US wildland fuels and their dependence on combustion efficiency, Atmos. Chem. Phys., 19, 1013-1026, https://doi.org/10.5194/acp-19-1013-2019, 2019.

Reisen, F., Meyer, C. P., Weston, C. J., and Volkova, L: Ground-Based field measurements of $PM_{2.5}$ emission factors from flaming and smoldering combustion in eucalypt forests, J. Geophys. Res-Atmos., 123, 8301-8314, https://doi.org/10.1029/2018JD028488, 2018.

Yokelson, R. J., Burling, I. R., Gilman, J. B., Warneke, C., Stockwell, C. E., de Gouw, J., Akagi, S. K., Urbanski, S. P., Veres, P., Roberts, J. M., Kuster, W. C., Reardon, J., Griffith, D. W. T., Johnson, T. J., Hosseini, S., Miller, J.W., Cocker III, D. R., Jung, H., and Weise, D. R.: Coupling field and laboratory measurements to estimate the emission factors of identified and unidentified trace gases for prescribed fires, Atmos. Chem. Phys., 13, 89–116, doi:10.5194/acp-13-89-2013, 2013a.

**R20.** Page 12, line 12-13. "Figure 5 shows a moderate increasing trend in the SSA at 870 nm, but no significant trend in the SSA at 401 nm." Please state how you checked for statistically significant trend.

**A20.** We've added the uncertainty in the slopes to the figure. The slope is only larger than the uncertainty for the 870 nm data (the longer time series).

**R21.** Page 12, line 29. "smoke was mostly sourced from a local fire (Rice Ridge)." How far was the fire? Can you estimate the smoke age?

**A21.** We added an estimated range of hours after the fire name in parentheses: "smoke was mostly sourced from a local fire (Rice Ridge) and about 2-4 hours old.

**R22.** Page 12, line 29. "Our peak-integrated proxy for particle size (4.02, smaller particle size)" Please describe the "peak-integrated proxy for particle size" in Section 2. Figure 6 (case study). Please add a second panel with high-resolution excess mixing ratios (BC/CO, PM2.5/CO, scattering/CO, trace gases/CO) so that the reader can compare the two peaks.

**A22.** Is the first part a suggestion to move the proxy to experimental section? We'd like to keep it in results since it is not a standard product. We've added most of the higher resolution data that has reasonable signal/noise to Fig, 6; subject to the caveats discussed above.

**R23.** Page 13, Section 3.6 Diurnal Cycles. I would expect diurnal cycle to be important for near-fire measurements due to diurnal variation in the emissions (e.g. Saide et al., 2015), oxidation and dilution. However, I would not expect much difference in aged regional smoke, whether it is observed during morning or evening hours. Here, focusing on extensive properties (PM2.5, BC, CO) is problematic as they depend mostly on dilution. I wonder if the diurnal cycle in Figure 7 has a small increase in morning only because more fresh plumes happened to reach the measurement site during morning hours. I recommend removing this section or concentrating on fresh plumes (e.g. CO > 0.5 or 1 ppm) and intensive properties (excess mixing ratios).

**A23.** We understand that multiple factors influence the diurnal profiles. Nevertheless, we think they are useful on several levels. They provide a relaxed, averaged case for model evaluation compared to strict point by point agreement in real time. Curiosity about the diurnal profiles reflecting real-time partitioning and general curiosity are some of the first questions we had and the diurnal cycles characterize the typical regional impacts even if the underlying reasons are not completely clear. Also our loose association of BC in evening and BrC in AM is probably relevant for a "typical "source to Missoula" delay. Our response to Referee #3 further develops the potential applications of our data.

**R24.** Page 14, line 11-13. "Despite our lower BC/CO ratio our PM/CO ratio was about half that measured in fresh smoke from aircraft. This suggests that OA evaporation, at least near the surface, may typically reduce PM air quality impacts on the time scale of several days." I do not think you can draw such a straightforward conclusion, as PM/CO ratio decreases with decreasing BC/CO. If both fuel and BC/CO are equal, then a lower PM/CO in aged smoke would suggest primary aerosol evaporation. Please check also abstract (page 1, line 18-22).

**A24.** We addressed part of this above. The broader conclusion comes from considering all available data for wildfires on page 10. We see that PM/CO dropped after aging on the Rim Fire (Forrister et al) to a value similar to ours, but not in Collier et al further north and higher altitude. In response to referee #1 we noted that a similar evaporation of PM was observed for a prescribed fire in a coniferous ecosystem. We agree we need to revise the text for people who may read only the conclusions and did not see on page 10 that POA volatility might vary by fuel type, the G-1 flights were further north than the Rim Fire, and that higher ambient temperature for smoke aging, as opposed to aging in general, may increase smoke evaporation rates.

Old text: "Despite our lower BC/CO ratio our PM/CO ratio was about half that measured in fresh smoke from aircraft. This suggests that OA evaporation, at least near the surface, may typically reduce PM air quality impacts on the time scale of several days."

New text: P14, L11: Despite our lower $\Delta BC/\Delta CO$ ratio our $\Delta PM/\Delta CO$ ratio was about half that measured in fresh smoke from aircraft. Taken together with aircraft measurements in aged wildfire smoke, this suggests that OA evaporation at higher ambient temperatures nearer the surface may typically reduce PM air quality impacts on the time scale of several hours to days."

**R25.** It seems that all linear fits are calculated with ordinary least squares method, which assumes that there is no uncertainty in x-direction. At least for Figs. 2, 3 and S1 a bivariate method would be more appropriate (see e.g. Cantrell et al., 2008).

**A25.** The requirement for linear regression is not quite as strict as "zero" uncertainty in the x value (a case which may not exist) and the rigorous requirement is perhaps summarized in simple terms a bit closer to ~ "linear regression is most accurate when $\Delta X$ is significantly smaller than $\Delta Y$." We did switch to orthogonal regression for Figure 2, which is updated and has a slightly changed slope. Orthogonal regression was not satisfying for Figure 3. The BC/PM plot had a visually inappropriate fit that weighted a single high value too much and gave an unrealistic intercept that was much larger than the near zero value clearly implied by a glimpse at the data. The effect on the slope was about a 20% reduction, but we elected to keep the linear regression figure here and in 4b and the supplement; in all cases the x-error is smaller than y-error.

**R26.** Please combine Tables 1 and 5 to avoid repetition. Please also check that you have defined the values in parenthesis in all Table captions. Is the study average a mean of enhancement ratios defined for each plume?

**A26.** We planned to do this, but ended up adding to Table 5 (per Referee #1) and electing to keep it separate.

References

Cantrell, C. A.: Technical Note: Review of methods for linear least-squares fitting of data and application to atmospheric chemistry problems, Atmos. Chem. Phys., 8(17), 5477–5487, doi:10.5194/acp-8-5477-2008, 2008.

Saide, P. E., Peterson, D. A., da Silva, A., Anderson, B., Ziemba, L. D., Diskin, G., Sachse, G., Hair, J., Butler, C., Fenn, M., Jimenez, J. L., Campuzano-Jost, P., Perring, A. E., Schwarz, J. P., Markovic, M. Z., Russell, P., Redemann, J., Shinozuka, Y., Streets, D. G., Yan, F., Dibb, J., Yokelson, R., Toon, O. B., Hyer, E. and Carmichael, G. R.: Revealing important nocturnal and day-to-day variations in fire smoke emissions through a multiplatform inversion, Geophysical Research Letters, 42(9), 2015GL063737, doi:10.1002/2015GL063737, 2015.

Vakkari, V., Beukes, J. P., Dal Maso, M., Aurela, M., Josipovic, M. and van Zyl, P. G.: Major secondary aerosol formation in southern African open biomass burning plumes, Nature Geosci., 11, 580–583, doi:10.1038/s41561-018-0170-0, 2018.

---

## Author Comment (AC3) · 25 Feb 2019

Response to Referee #3

We thank the Referee for all their comments, which have helped improve the paper as described below. The Referee suggestions are shown in full along with our detailed response/revisions in an "R#, A#" format next.

**R1.** This manuscript presents a major wildfire aged smoke measurement of some aerosol properties and trace gases in Missoula (US) during August-September 2017. During this period the measurement location was affected by several smoke plumes from wild fires, more importantly a smoldering and nighttime fire chemistry case is presented. Model back trajectories and satellite retrievals allowed for some of the fire locations to be identified and investigated. In summary, this data set presented here contains approx. 500 h of ground-based plume measurements and can provide valuable information on statistics for modeling and emission factors based on flaming vs. smoldering combustion on a regional scale. The prescribed burning comparisons are an interesting start to a much-needed solution. I think this paper is acceptable but could benefit from a deeper look into the implications for modeling use via smoldering and nighttime chemistry.

**A1.** Referee #3 shares our desire for more insight into flaming vs smoldering and day vs night chemistry as evidenced by the comment above and several below. We therefore discuss this goal in detailed context at the outset of this response. Even in a lab where fire emissions mix with a constant background, once the flame front moves, flaming and smoldering are mixed. Finding the separate contributions requires a mathematical analysis such as in Yokelson et al., (1996). Even that is approximate because the relative contribution of pyrolysis and glowing to smoldering can vary over time and space, and both processes are themselves a complex mix. Sekimoto et al., (2018) show how the pyrolysis itself can be broken down into two complex factors. In the field, a real fire can mix with multiple different layers of the atmosphere or other fires during transport, which can distort some signatures of flaming vs smoldering as discussed in detail in Yokelson et al., (2013b). One scenario that is not uncommon is smoke traveling slowly at low altitude from nearby fires being older and initially stratified from smoke above it that traveled faster from fires further away. This can be followed by vertical mixing that blends smoke of different ages from different fires at some distance from the sources. MCE is a pretty good rough indicator of flaming vs smoldering (F/S) if no mixing effects distort it as discussed in Yokelson et al., (2013b). BC/CO can also be used as an F/S indicator and it should be preserved with less distortion if mixing only occurs with background since BC is rare in background air unlike $CO_2$. If $BC/CO_2$ was constant for flaming then BC/CO would be essentially a proxy for $CO_2/CO$ or MCE by rearrangement. However, $BC/CO_2$ can vary a lot for flames perhaps mostly because turbulence in diffusion flames has a small effect on the $CO_2$ yield but a much larger effect on the BC yield (Shaddix et al., 1994). In a near-field study of fires there is some chance to resolve flaming vs smoldering or day vs night differences. In addition, most prescribed fires are less than a day long and most of the smoke is lofted in a way that is accessible to airborne sampling. However, wildfires can burn 24/7 for months with dynamic/shifting dispersion scenarios that may be accompanied by changes in emissions chemistry. Thus, it is difficult to assess how well the emissions sampled from any platform represented the overall fire output (Yates et al., 2016; Saide et al., 2015). In this study we monitor smoke mixtures at a distance and we are not best positioned to separately characterize pure flaming and smoldering or pure night and day chemistry. However, we can measure the net integrated downwind impact of a

huge number of regional fires, including mixing. This provides an opportunity to check if our observations of conserved tracers are consistent with the data being used to represent wildfire sources in models. I.e. the data can help evaluate measurements, emissions inventories, and models. Comparisons are possible to our exact time series or diurnal cycles for a more relaxed test. Also, for example is BC/CO at a heavily impacted surface site generally consistent with BC/CO in the emissions inventories that serve as model input, or do our results suggest some changes are worth considering? We also provide actual values of dynamic ratios (e.g. PM/CO) that can help elucidate the nature of plume evolution. We have reached out to several modeling groups interested to compare their model output to our "ground truth." We've also recently joined collaborative efforts to institute ground-based near-field sampling as an approach to sample a greater fraction of the total output from wildfires than can be done from the air alone. Modeling, near-field and downwind airborne sampling as well as ground-based sampling at various altitudes (e.g. surface through mountain-tops) all have a key role to play.

Yokelson, R. J., Andreae, M. O., and Akagi, S. K.: Pitfalls with the use of enhancement ratios or normalized excess mixing ratios measured in plumes to characterize pollution sources and aging, Atmos. Meas. Tech., 6, 2155-2158, doi:10.5194/amt-6-2155-2013, 2013b.

Saide, P. E., Peterson, D., da Silva, A., Anderson, B., Ziemba, L. D., Diskin, G., Sachse, G., Hair, J., Butler, C., Fenn, M., Jimenez, J. L., Campuzano-Jost, O., Perring, A., Schwarz, J., Markovic, M. Z., Russell, P., Redemann, J., Shinozuka, Y., Streets, D. G., Yan, F., Dibb, J., Yokelson, R., Toon, O. B., Hyer, E., and Carmichael, G. R.: Revealing important nocturnal and day-to-day variations in fire smoke emissions through a multiplatform inversion, Geophys. Res. Lett., 42, 3609-3618, doi:10.1002/2015GL063737, 2015.

Sekimoto, K., Koss, A. R., Gilman, J. B., Selimovic, V., Coggon, M. M., Zarzana, K. J., Yuan, B., Lerner, B. M., Brown, S. S., Warneke, C., Yokelson, R. J., Roberts, J. M., and de Gouw, J.: High- and low-temperature pyrolysis profiles describe volatile organic compound emissions from western US wildfire fuels, Atmos. Chem. Phys., 18, 9263-9281, https://doi.org/10.5194/acp-18-9263-2018, 2018.

We've modified text in various places as described in response to more detailed comments below:

Major comments

**R2.** Page 3 line 15: The author indicates that this can be used to inform model mechanisms; however, outside of presenting numbers for ratios (which can and is helpful) without context of in what way to use these ratios. Meaning, all numbers are not created equal, in what modeling scenario should these new numbers or measurements be applicable?

**A2.** We agree with Referee that more than three words are valuable here early on in the paper to summarize the value and potential applications of our data and made the following change:

P3, L14: truncate the sentence by deleting ", which can be compared to changes in aerosol optical properties and inform model mechanisms." Add new text before "We present…"

The main goals of this work are to document the net, combined effect of numerous fires from a heavily impacted surface site embedded in the region and thus, also help assess the

representativeness of field measurements, emissions inventories, and models. In more detail; we characterize the smoke impacts on a population center and we document the real-world regional significance of brown carbon. Comparisons are possible to our time series of BC, CO, PM, etc or diurnal cycles for these species for a more relaxed test. Our real-time through study-average ratios for "inert" tracers such as $\Delta BC/\Delta CO$ are compared with $\Delta BC/\Delta CO$ in the field measurements that are available to build emissions inventories that serve as model input. The time-resolved and study-average values of dynamic ratios (e.g. $\Delta PM/\Delta CO$) help elucidate the net effect of secondary aerosol formation and evaporation. Our measurements provide real-world aerosol optical properties (e.g., SSA, AAE, etc.) and can be used with the aerosol mass data at real-time through study-average resolution to probe multi-step, bottom-up calculations of climate-relevant aerosol optical properties.

**R3.** Are these numbers for nighttime generated smoke? Can one use these numbers when a fire is detected at night or during the day and expected to be smoldering? E.g. page 6 line 5: "time series of mixing ratios" is helpful to point out in detail. E.g. BC/CO as a function of distance would be helpful.

**A3.** While we can't measure pure night-time emissions (see above), the text here needed to be rephrased to clarify that time series of multiple data types, ratios and other parameters are useful.

P, L4, old text: "We converted the time series of mixing ratios for each analyte measured into a form that is broadly useful to others for implementation in local to global chemistry and climate models. To do this, we produce emission ratios (ERs) and enhancement ratios."

New text: "Time series are useful to characterize impacts and evaluate models, but we also used the time series of mixing ratios or concentrations for each analyte measured to derive other values that are broadly useful for study comparisons and implementation in local to global chemistry and climate models. As part of this, we produced emission ratios (ERs) and enhancement ratios."

**R4.** Page 4 line 3-5; brief discussion of the uncertainties; there needs to be more in this paper about those uncertainties associated with each calculation and its use in a modeling platform or intended use.

**A4.** Referee 1 and 2 also shared this concern and we agreed. Error bars and uncertainties in slope were added to figures and the error discussion was expanded in the text. We hope the improvements described in detail in those responses will address the concerns of Referee 3 also.

**R5.** Page 6, line 18-21 MCE is not a good indicator of flaming vs smoldering compared to BC and CH4 ratios to CO, needs a citation, unless you are planning on providing evidence in this paper of this using the data collected?

**A5.** What we mean is MCE can be distorted at a distance as discussed above and at length in Yokelson et al., (2013). We have added text to clarify that we meant MCE is distorted *in this particular study*.

P6, L20: We added "in this study" before "as in measurements…" and the citation to Yokelson et al 2013b.

**R6.** Page 7, line 18-27 it seems that the authors had an opportunity with this data set to take a look into the various composition of fuels and impacts on transported chemistry. The small caveat to this is that hysplit will not likely give you 100% certainty on the origin, but with the fires that were identified, I would have liked to see an attempt to separate out measured emissions vs fuel types. This could potentially be a nice case study for Lolo Peak fire and Rice Ridge fire. As this fuels composition could be one explanation of the presented results differences between the other studies.

**A6.** This would be nice, but both nearby fires burned in complex mixed-coniferous ecosystems that had a strong variation in vegetation mix with altitude. The back-trajectories have limited vertical resolution and fuel consumption weighting by component varied with time in unknown ways. Thus, while the goal is worthwhile we feel it is best addressed in a near-field study. We made a text change to clarify the general probable lack of pure sources.

P7, L25, old text: "Many of the longer smoke impacts that spanned several days were necessarily integrated as a single event for calculating ratios between species, but also probed as smaller "sub-events" to explore their source attribution, which could be mixed (Tab. S1)."

New text: "Many of the longer smoke impacts that spanned several days were necessarily integrated as a single event for calculating ratios between species, but we also initialized back trajectories from local maxima to further explore the source region of the smoke, which was probably always mixed to some extent (Tab. S1)."

**R7.** Page 8, line 17 "time since emission" I would have like a deeper dig into this as the results all hinge upon the accuracy of this. The authors claim the smoke came from late afternoon to nighttime but do not show this anywhere outside of the supplemental materials. And since hysplit does not include full chemistry it seems odd to use it to look at full chemistry transported, but as you indicated the ratios compared to the relatively conserved CO should be okay.

**A7.** What we meant was, in general, smoke may have a greater transport age or time since emission than may be indicated by a "photochemical age". This can always occur, but is perhaps most likely for wildfires which tend to blow up late in the day.

P8, L17 now reads: "However, the "time since emission" is potentially longer than indicated by a "photochemical age" since,"

**R8.** Page 8, line 35 the separation of smoldering vs flaming vs residual smoldering is difficult, particularly in modeling and source attribution. If there was a ratio or tracer method that was found to actually indicate one of the other this was not clear to me reading this. It appears the distinction was made based off time of day (and one case presented grew at night), knowledge of fires state, and measured chemistry. Which is nice but going forward most cases wont have all that information.

**A8.** We don't fully understand this comment, but our point on P8, L35 was, for one example, a measurement of furan/CO from a different study measuring initial emissions close to a fire source could be used with our CO data to estimate the initial furan for a model simulation.

P8, L36: We changed "when emission ratios to CO" to "if those gases emission ratios to CO"

**R9.** Page 9, line 17. It appears that this study used only three heights to initialize hysplit, but did not indicate why those heights where chosen (if it was based purely on the elevation of the terrain then that makes sense). However, it does not include the effects of plume rise? As smoldering smoke tends to pool near the surface but can reach higher elevations, and vice versa for flaming smoke.

**A9.** The heights for back trajectories roughly indicate the following: 500 m AGL (height of frequently-observed elevated morning smoke layers that then mixed down into the Missoula valley at circa 11 AM to cause a mid-day PM peak); 3000 m AGL (common injection altitude for wildfires, e.g. assume maximum possible transport at injection altitude before mixing down), 1200 m AGL (intermediate point). In retrospect a lower starting elevation near 50-100 m AGL could also be useful, but the accuracy would likely be lower. Valley flows, up/downslope, and local vertical mixing are difficult to model in complex terrain. We often don't know if smoke arrived at ground level or mixed down and wind direction varies with altitude, so we initialize the back trajectories at several heights to generate possibilities. The sum of all the exploratory back trajectories is consistent with complex, but impressive regional coverage

[Figure]

**R10.** Consider the references

Wilkins JL, Pouliot G, Foley K, Appel W, Pierce T (2018) The impact of US wildland fires on ozone and particulate matter: a comparison of measurements and CMAQ model predictions from 2008 to 2012. International Journal of Wildland Fire, https://doi.org/10.1071/WF18053.

Zhou L, Baker KR, Napelenok SL, Pouliot G, Elleman R, O'Neill SM, Urbanski SP, Wong DC (2018) Modeling crop residue burning experiments to evaluate smoke emissions and plume

transport. Science of the Total Environment 627, 523-533, https://doi.org/10.1016/j.scitotenv.2018.01.237.

**A10.** These are both good examples of modeling and impacts as we added the citations on P1, L37.

**R11.** Page 9, line 33 aging and/or higher average temperatures at lower elevation may encourage some OA evaporation and reduce downwind PM impacts. This line is very interesting and should be expanded upon, as it's a critical finding from this study. What here is indicated as higher average temperatures? Is this flaming stage or just hot temperatures in the atmosphere as the plume ages? (page 10, line 12-15 also are confusing for the same reason "and thus strongly cooling"). Furthermore, can a statement be made in this section about smoldering plumes traveling in hotter temperatures or temperature of plume on evaporation of PM? This point would be good to attempt to relate to prescribed burns, as the emissions tend to be more toxic (or higher for PM) from the incomplete combustion and lower temperatures of burns and therefore longer smoldering time periods.

**A11.** Because temperature tends to decrease with altitude, smoke transported closer to the surface, or that mixes down, may experience higher ambient temperature, which could drive enhanced evaporation compared to measurements made higher in atmosphere or at high surface elevations. This comment reminded us that higher PM in early AM could have some contribution from gas-particle partitioning. We don't address relative toxicity of smoke from PF and WF, but note that PF are typically designed to have less smoldering than wildfires.

Changes:

P9, L33: We added "ambient" after "higher average"

P10, L10: we changed "some net evaporation of PM is occurring between the wildfire sources and our surface site." To "some net evaporation of PM is occurring at lower, warmer altitudes during transport between the wildfire sources and our surface site."

**R12.** Also, for the section 3.2 (page 10, line 3-5) are the authors discussing BC on average or BC for smoldering cases. It seems from the way its written that this ratio is for smoldering and the one presented in Liu et al. is for flaming? Could there be a statement made such as BC/PM < x is expected to be from smoldering while BC/PM > x is expected to be flaming?

**A12.** P10, L3 & 4: we added "study-" before "average" in two locations to clarify. We don't have a great lab data set for wildfire fuels for BC/PM as a function of MCE and in our downwind study BC/PM can be altered by PM evolution. BC/PM initial emissions are also variable as discussed above and explored in other responses.

**R13.** Page 13, line 20 It states that a possibly explanation is that more BC is being generated during the day, however it transported to the site overnight in order to arrive by 5am. Or is this statement meant to mean, the transported plume that remained over Missoula cooked during the daytime hours and generated more BC during the daytime while at Missoula?

**A13.** In-situ BC generation is not possible and time delays between emission and arrival in Missoula vary. Our thought was that more BC may be generated by increased flaming during the

day at the fire sources less than several hours upwind and that signal could survive and could contribute to higher (less diluted) levels in general in an evening peak.

P13, L20, old text: "One possible explanation for this is that despite variation in mixed layer height there is "typically" an increase in the flaming to smoldering ratio that produces more black carbon during the day. "

New text: "One possible explanation for this is that despite variation in mixed layer height there is "typically" an increase in the flaming to smoldering ratio that produces more black carbon and less brown carbon during the day. If nearby (less diluted) fires with shorter transport times strongly influence the peak times a signal of diurnal variation at the source could be partially evident at our site. "

Minor comments

**R14.** There is a need for a careful defining of terms. Some terms are used before they are defined, and others are never defined. And I believe all terms should be defined that are used in the abstract. E.g. BrC is used on page 1 line 23 and defined later on line 28; "US" is used on page 1 line 37 and is not defined. The authors need to decide whether or not to abbreviate which terms and remain consistent, e.g. Biomass burning appears as BB sometimes and other times not, also Air quality is sometimes AQ.

**A14.** We proofread and tied to eliminate the errors.

**R15.** A through grammar check is needed. There are some run on sentences and some missed placed commas and periods. E.g. page 2 line 3-10 very long run-ons.

**A15.** We proofread and tied to eliminate the errors.

**R16.** Page 10, line 35 does this ratio come with a trend or can expect numbers be inferred?

**A16.** What we meant was that even though our smoke was aged, BrC was still important. That implied that aging decreases BrC, which may not be obvious.

P10, L35: We changed "in our moderately aged smoke." to "despite some aging of the smoke at our site."

**R17.** Page 11, line 36 what is meant by "870 nm is unity to a good approximation " the transitions at the end of paragraphs in my opinion are not needed (e.g. Page 13, line 12) " which we examine next"

**A17.** We changed "unity" to "one" and deleted some transitions.